# Soil bacterial networks are less stable under drought than fungal networks

Franciska T. de Vries [1], Rob I. Griffiths [2], Mark Bailey[2], Hayley Craig[1], Mariangela Girlanda[3,4], Hyun Soon Gweon [2], Sara Hallin [5], Aurore Kaisermann[1], Aidan M. Keith [6], Marina Kretzschmar[5], Philippe Lemanceau[7], Erica Lumini [4], Kelly E. Mason [6], Anna Oliver[2], Nick Ostle[8], James I. Prosser [9], Cecile Thion [9], Bruce Thomson[2] & Richard D. Bardgett [1]

Soil microbial communities play a crucial role in ecosystem functioning, but it is unknown how co-occurrence networks within these communities respond to disturbances such as climate extremes. This represents an important knowledge gap because changes in microbial networks could have implications for their functioning and vulnerability to future disturbances. Here, we show in grassland mesocosms that drought promotes destabilising properties in soil bacterial, but not fungal, co-occurrence networks, and that changes in bacterial communities link more strongly to soil functioning during recovery than do changes in fungal communities. Moreover, we reveal that drought has a prolonged effect on bacterial communities and their co-occurrence networks via changes in vegetation composition and resultant reductions in soil moisture. Our results provide new insight in the mechanisms through which drought alters soil microbial communities with potential long-term consequences, including future plant community composition and the ability of aboveground and belowground communities to withstand future disturbances.

[1] School of Earth and Environmental Sciences, The University of Manchester, Oxford Road, Manchester M13 9PT, UK. [2] Centre for Ecology & Hydrology Wallingford, Maclean Building, Benson Lane, Crowmarsh Gifford, Wallingford, Oxfordshire OX10 8BB, UK. [3] Department of Life Sciences and Systems Biology, University of Torino, Viale Mattioli 25, 10125 Torino, Italy. [4] CNR—Institute for Sustainable Plant Protection, UOS Turin, Viale Mattioli 25, 10125 Torino, Italy. [5] Department of Forest Mycology and Plant Pathology, Swedish University of Agricultural Sciences, 750 07 Uppsala, Sweden. [6] Centre for Ecology & Hydrology Lancaster, Library Avenue, Bailrigg, Lancaster LA1 4AP, UK. [7] Agroécologie, AgroSup Dijon, INRA, Univ. Bourgogne Franche-Comté, F-21000 Dijon, France. [8] Lancaster Environment Centre, Lancaster University, Library Avenue, Bailrigg, Lancaster LA1 4AP, UK. [9] School of Biological Sciences, University of Aberdeen, Cruickshank Building, St Machar Drive, Aberdeen AB24 3UU, UK. Correspondence and requests for materials should be addressed to F.T.d.V. (email: franciska.devries@manchester.ac.uk)

Soils harbour highly diverse microbial communities that are of crucial importance for soil functioning[1,2]. A major challenge is to understand how these complex microbial communities respond to and recover from disturbances, such as climate extremes, which are predicted to increase in frequency and intensity with climate change[3]. Many studies have demonstrated that climate extremes, such as drought, can have considerable effects on soil microbial communities, often with consequences for ecosystem processes and plant community dynamics[4–6]. It has also been shown that different components of the microbial community respond differently to drought, with soil fungi being generally more resistant, but less resilient, than bacteria[4,7–9]. Moreover, the recovery of fungi and bacteria from drought is differentially governed by plant physiological responses to drought[10], for example by reducing the transfer of recently plant-assimilated C to bacteria, but not to fungi[11]. Although these past studies provide important insights into the impacts of climate extremes and other disturbances on soil microbial communities, they have almost exclusively focussed on single properties of soil microbial communities and their functioning, rather than on the multitude of direct and indirect interactions that occur between the networks of microbial taxa that co-exist in soil. This represents an important gap in understanding, given that climate extremes not only have the potential to reorganise networks of interactions between co-existing soil microbial taxa[12], but also the properties of these networks themselves could determine their response to disturbances[1,9,13].

Evidence is mounting that properties of ecological networks, which might represent interactions between co-existing organisms, can influence the response of communities to environmental change, including climate extremes[1,9,14]. Theoretical studies, for example, predict that ecological networks that consist of weak interactions are more stable than those with strong interactions[15,16], and that compartmentalisation and presence of negative interactions increase the stability of networks under disturbances[16–18]. Further, modelling studies show that increasing strength of a few key interactions within a food web can destabilise trophic cascades[14]. Thus, communities in which a large proportion of members are connected through positive links are deemed to be unstable; in such communities, members may respond in tandem to environmental fluctuations, resulting in positive feedback and co-oscillation[16]. Negative links might stabilise co-oscillation in communities and promote stability of networks[16]. Despite this knowledge, and an increasing use of network analysis in ecology[1,19–21], our understanding of co-occurrences or potential interactions within complex soil microbial communities, and how they respond to and recover from disturbances such as climate extremes, remains scant[2,12].

Here, we experimentally investigated how prolonged summer drought impacts soil fungal and bacterial networks, and whether this response is consistent with the properties of these networks. We also aimed to test the role of plant communities in the response of bacterial and fungal networks and communities to drought, and what the implications are for soil functioning. Because soil bacterial communities are less resistant, but more resilient (i.e., show stronger fluctuations over time), to drought than fungal communities, we expected bacterial networks to display more destabilising properties. First, we expected correlations between bacterial OTUs to be stronger overall than those between fungal OTUs, and that a larger proportion of these correlations are negative in fungal networks. We tested these assumptions using all possible correlations in bacterial and fungal networks separately, thus including weak and non-significant interactions. Second, we expected a larger proportion of bacterial than fungal OTUs to be significantly correlated in co-occurrence networks, and these networks to be more connected for bacteria

than for fungi. Third, we expected networks of positive correlations to have lower modularity for bacteria than for fungi. We tested these two latter assumptions using co-occurrence networks that only included significant, positive correlations. Positive correlations within fungal and bacterial communities can represent a range of interactions or simply indicate that they are responding in the same way to a change in environmental conditions[21,22]. Thus, although caution is needed in interpreting them[22,23], co-occurrence networks can reveal information on co-oscillation of microbial taxa[21] and the stability of communities. Fourth, we expected drought-induced changes in plant growth and community composition to be associated with the trajectory of recovery of microbial networks and communities, and this link to be stronger for fungal communities, with these being the first users of belowground plant C inputs[24]. Finally, given evidence for links between soil microbial communities and biogeochemical cycles[2], we expected drought-induced changes in both fungal and bacterial communities and networks to influence soil functioning.

The above expectations were tested using a field-based mesocosm experiment consisting of plant communities varying in relative abundance of four common grassland species with contrasting life history strategies: a fast-growing, resource exploitative grass (*Dactylis glomerata*) and herb (*Rumex acetosa*), and a slow-growing, resource conservative grass (*Anthoxanthum odoratum*) and herb (*Leontodon hispidus*). Plant communities were dominated by one of these species and varied in evenness, such that each plant species dominated a low and medium evenness community in which the three other species had equal abundances. In addition, a high evenness treatment was included in which all species had equal abundances (see Methods and Supplementary Table 1). These treatments were intended to create a gradient of drought-induced shifts in plant communities in response to a summer drought that was imposed in the second growing season. We then quantified responses of plant communities, soil bacterial and fungal communities and their overall and co-occurrence networks, and fluxes of greenhouse gases ($CO_2$ and $N_2O$; products of plant and microbial respiration and the microbial processes nitrification and denitrification) over time in response to drought.

Our results show that bacterial and fungal networks differ significantly in key properties that might inform on their stability under disturbance: bacterial networks show more destabilising properties than fungal networks, and these properties are stimulated by drought. We also show that a drought-induced shift to dominance of the fast-growing grass has long-lasting direct and indirect legacy effects on bacterial networks and communities, with the potential to reinforce changes in plant community composition and affect aboveground and belowground responses to future disturbances.

## Results

**Bacterial and fungal communities.** Both fungal and bacterial communities were strongly, but contrastingly, affected by drought (Fig. 1). Fungal richness and evenness increased during the drought period, but rapidly recovered to control levels 1 week after rewetting. In contrast, drought decreased bacterial richness and evenness; this effect was strongest 1 week after rewetting and persisted for 2 months. We used Bray–Curtis similarity as a measure of resilience of bacterial and fungal communities. When similarities between drought and control communities were significantly lower than those before drought, communities were assumed to be affected by drought and had not recovered. When similarities did not differ significantly, the communities were assumed to be unaffected or had recovered[25]. We found that similarities between bacterial drought and control communities

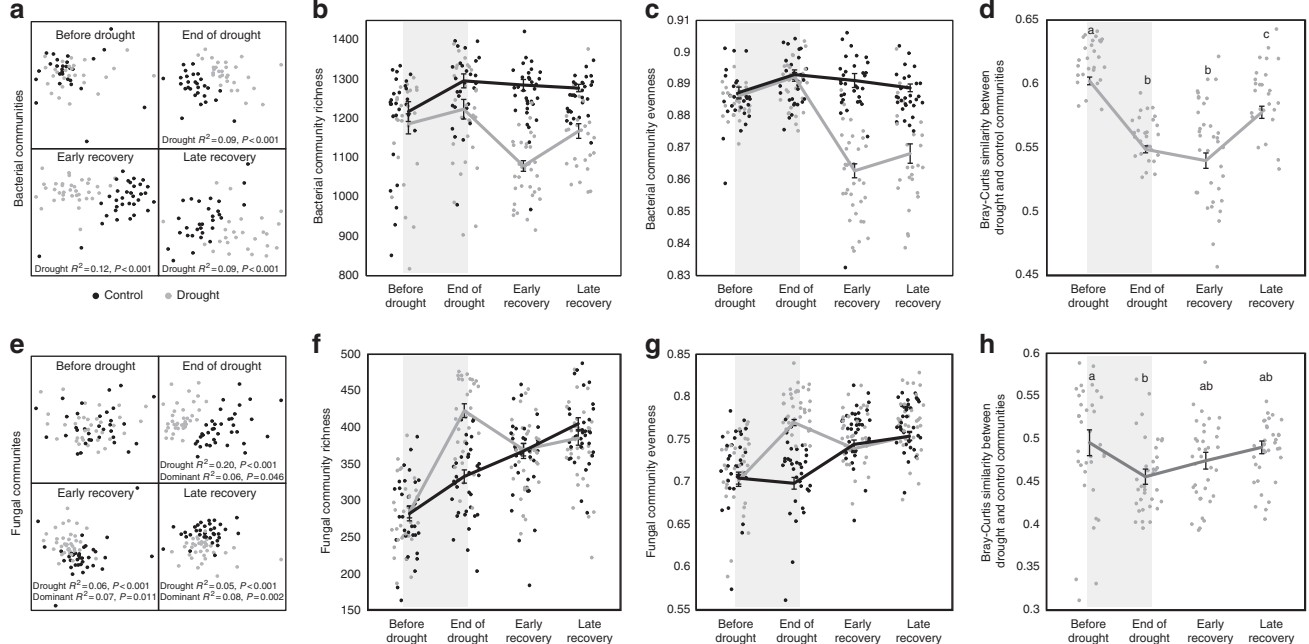

**Fig. 1** Bacterial and fungal community response to drought. The duration of the drought is indicated as a shaded area in **b**–**d** and **f**–**h**. NMDS of bacterial (**a**) and fungal (**e**) community composition shows that drought significantly affected bacterial and fungal communities at all sampling dates following drought (ADONIS). Bacterial (**b**, **c**) and fungal (**f**, **g**) richness and evenness were strongly affected by drought, but this response differed over time (Repeated measures ANOVA sampling × Drought interaction $F_{3,153} = 10.4$, $P < 0.001$ and $F_{3,195} = 14.8$, $P < 0.001$ for richness of bacteria and fungi, respectively; Sampling × Drought interaction $F_{3,153} = 23.1$, $P < 0.001$ and $F_{3,195} = 23.1$, $P < 0.001$ for evenness of bacteria and fungi, respectively). The similarity between drought and control communities changed over time for both bacteria and fungi, but bacterial drought and control communities were still less similar than before drought at late recovery (ANOVA sampling effect $F_{3,80} = 40.9$, $P < 0.001$ and $F_{3,103} = 2.8$, $P = 0.04$ for bacteria and fungi, respectively). In **b**–**d** and **f**–**h**, dots represent individual observations and lines indicate means ± 1 SE. In **d**, **h**, symbols with the same letter are not statistically different

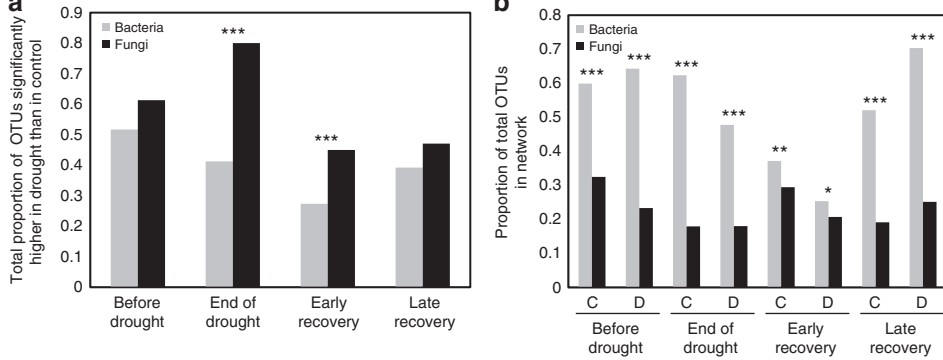

**Fig. 2** Relative abundance and proportion of total bacterial and fungal OTUs included in co-occurrence networks. A significantly larger proportion of fungal OTUs than bacterial OTUs increased under drought at the end of the drought and early recovery (**a** $\chi^2 = 122$, df $= 1$, $P < 0.001$ and $\chi^2 = 23.2$, df $= 1$, $P < 0.001$, respectively). A significantly larger proportion of bacterial OTUs was included in bacterial co-occurrence networks than of fungal OTUs in fungal co-occurrence networks (**b** $\chi^2$-tests for all drought (D) and control (C) treatments over time were significant). Asterisks indicate the significance of the difference between proportions in bacterial and fungal communities (*$P < 0.05$, **$P < 0.01$, ***$P < 0.001$)

were significantly lower than they were before drought at all post-drought samplings, indicating an ongoing effect of drought on those communities. In contrast, similarities between fungal drought and control communities were only lower than before drought at the end of the drought, and weakly so (Fig. 1). In addition, we also found that a larger proportion of fungal OTUs increased in relative abundance in response to drought than did bacterial OTUs (Fig. 2a and Supplementary Fig. 1).

We identified individual bacterial and fungal indicator OTUs that were both high in abundance and responded strongly in their abundance to drought (Methods, Supplementary Fig. 1). We also distinguished between drought-sensitive and drought-tolerant indicators, which respectively decreased and increased in response to drought. Both drought-tolerant and drought-sensitive bacterial indicators were mainly found within the phylum *Verrucomicrobia* and the class *Alphaproteobacteria*, and additional bacterial drought-sensitive indicators belonged to the phylum *Actinobacteria* (Supplementary Data 1). Fungal drought-tolerant indicators belonged to the phyla *Ascomycota* and *Glomeromycota* (Supplementary Data 2), whereas all fungal drought-sensitive indicators belonged to the phylum *Zygomycota*, specifically of the family *Mortierellaceae*.

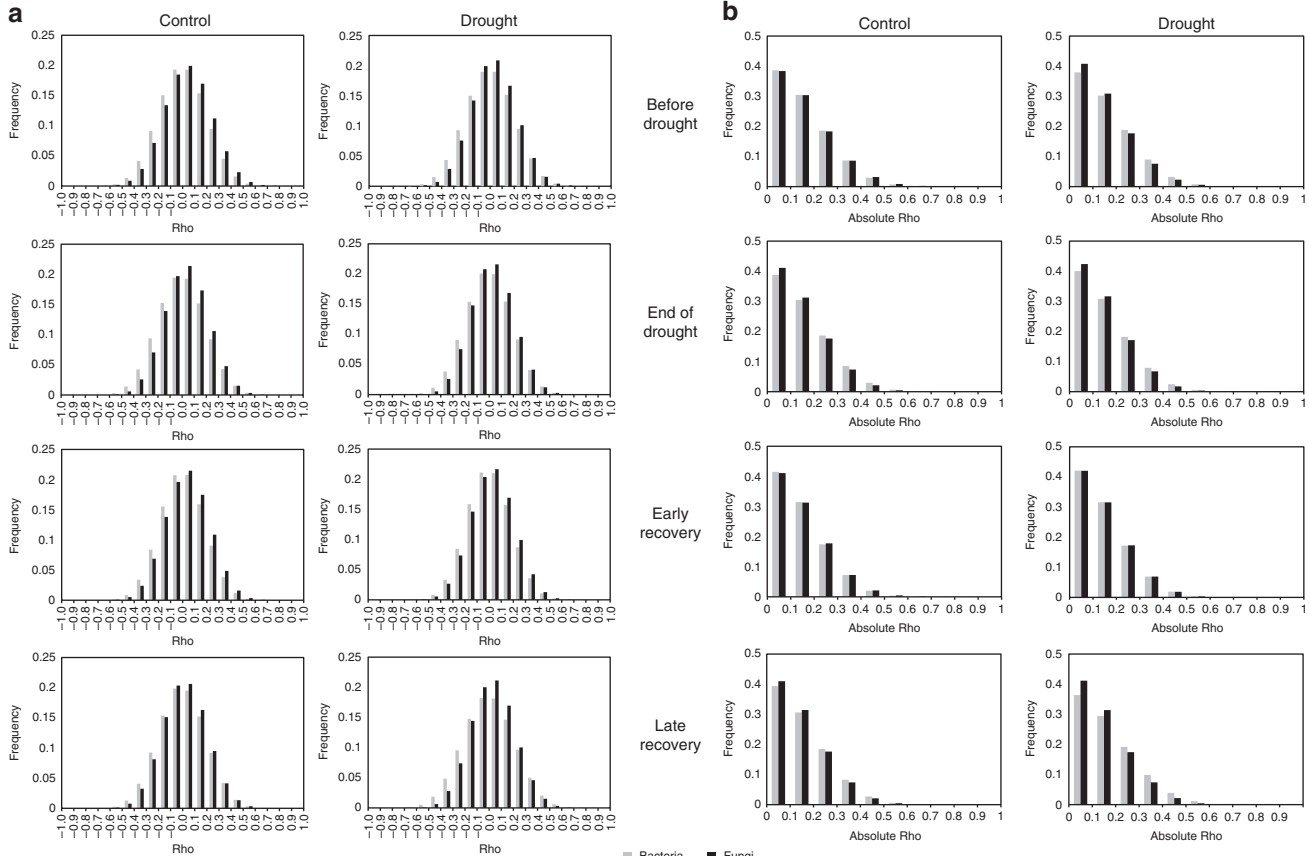

**Fig. 3** Frequency distributions of correlations in fungal and bacterial networks. Frequency distributions of the sign and strength of all correlations in fungal and bacterial networks (**a**), and of the absolute strength of all correlations in fungal and bacterial networks (**b**), in drought and control treatments over time (top to bottom). Correlations in bacterial networks are light grey, correlations in fungal networks are dark grey

**Bacterial and fungal networks**. When considering all correlations, the links between bacterial OTUs were consistently stronger than those between fungal OTUs (Fig. 3, ANOVA; $F_{1,18,778,692} = 5577$, $P < 0.0001$), supporting our expectation that potential interactions in bacterial networks are stronger than those of fungal networks. We also found that fungal networks consistently had fewer negative correlations than bacterial networks (Fig. 3, two-sided $\chi^2$-test of proportions; $\chi^2 > 500$ and $P < 0.0001$ for all fungal–bacterial control network pairs). When considering only significant correlations ($\rho > 0.6$, $P < 0.01$), again we found that fungal networks contained fewer negative correlations than bacterial networks (two-sided $\chi^2$-test of proportions; $\chi^2 > 10$ and $P < 0.001$ for all fungal–bacterial control network pairs). Drought reduced the proportion of negative correlations in bacterial networks at the end of the drought, and in fungal networks after 2 months of recovery (two-sided $\chi^2$-test of proportions; $\chi^2 = 9.6$, $P = 0.002$ and $\chi^2 = 9.5$, $P = 0.002$, respectively).

We then used these significant correlations to construct co-occurrence networks, consisting only of positive correlations. Bacterial co-occurrence networks were larger, more connected and less modular than fungal networks. A larger proportion of bacterial OTUs was included in co-occurrence networks than of fungal OTUs (Fig. 2b). The larger bacterial co-occurrence networks also had a higher node-normalised degree (the number of connections a node has standardised by the total number of connections in the network[26]) and betweenness (the number of paths through a node[26]) than fungal networks (Figs. 4, 5), while a lower clustering coefficient[26] indicated a marginally lower modularity in bacterial networks than in fungal networks (ANOVA; $F_{1,8} = 3.9$, $P = 0.08$, Supplementary Table 2).

Membership of bacterial networks was more constant over time than that of fungal networks, indicating consistent correlations between taxa (Supplementary Fig. 2). In addition, drought increased the connectedness and centrality of nodes in bacterial networks, while it decreased these properties in fungal networks (Fig. 5). When we analysed the combined co-occurrence networks, including both fungi and bacteria, networks were dominated by bacterial nodes and showed similar dynamics to bacteria-only networks (Supplementary Fig. 3 and Supplementary Table 2).

We found that indicator OTUs, regardless of whether they were drought-sensitive or drought-tolerant, were more connected than non-indicator OTUs in bacterial networks (Fig. 5), while in fungal networks drought-sensitive indicators were more central and connected than non-indicators. In particular, highly central in bacterial drought networks were drought-tolerant taxa of the genera *DA101* and *Candidatus xiphinematobacter* (*Verrucomicrobia*) and *Rhodoplanes* (*Alphaproteobacteria*) (Supplementary Data 1). We also found that the centrality (betweenness) and connectedness (normalised degree) of bacterial network OTUs was positively related to their relative abundance, but only in drought networks (regression; $R^2 = 0.016$, $P = 0.0002$, df = 884 for betweenness in drought late recovery network; $R^2 = 0.037$, $P = 0.0002$, df = 367; $R^2 = 0.010$, $P = 0.002$, df = 884, for normalised degree in drought early and late recovery networks). In contrast, these properties were strongly positively related to OTU relative abundance in fungal control networks. Before the simulated drought, connectedness was predicted by OTU abundance in both fungal control and drought networks (regression; $R^2 = 0.37$ and $P < 0.0001$, df = 86 and $R^2 = 0.22$

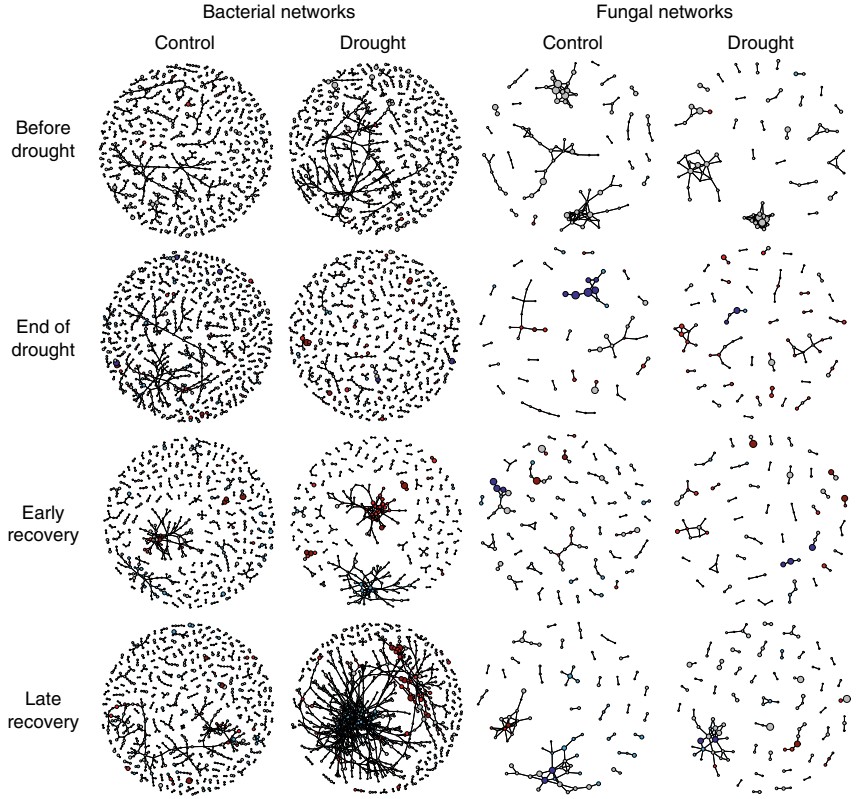

**Fig. 4** Bacterial and fungal co-occurrence networks over time as affected by drought. Nodes represent individual OTUs; edges represent significant positive Spearman correlations ($\rho > 0.6$, $P < 0.001$). Light blue and light red OTUs decrease and increase under drought, respectively; dark blue and dark red OTUs indicate high abundance OTUs that decrease and increase strongly under drought, respectively (drought-sensitive and drought-tolerant indicator OTUs). For detailed network properties, see Supplementary Table 1

and $P < 0.0001$, df = 111, respectively), and both centrality and connectedness were predicted by relative abundance in the late recovery control network (regression; $R^2 = 0.49$ and $P < 0.0001$, and $R^2 = 0.24$ and $P < 0.0001$, df = 101 for betweenness and normalised degree, respectively).

For most bacterial networks, there was a negative relationship between OTU relative abundance and the significance ($P$ value) of their response to drought. This indicates that the most abundant bacterial OTUs showed the strongest response to drought (regression; $R^2 > 0.01$ and $P < 0.001$ in end of drought and late recovery control and drought networks; $R^2 = 0.024$ and $P = 0.003$ in early recovery drought network); these relationships were absent in fungal networks.

**Links between plant and microbial community properties**. Drought caused a strong shift in plant community composition (Fig. 6a–d). Biomass of the fast-growing grass *D. glomerata* increased significantly relative to other plant species, and this increase was responsible for a divergence between control and drought plant communities (Fig. 6e, f). We assessed the extent to which this increased *D. glomerata* biomass was associated with the relative abundances of bacterial and fungal network OTUs during recovery from drought. We found more significant correlations of *D. glomerata* biomass with OTUs in bacterial networks, and these correlations were stronger than those with fungal network OTUs (Fig. 7). Moreover, in bacterial drought networks, the strength of the correlations with *D. glomerata* increased when nodes were more central and connected and might thus be potential keystone taxa[26] (Fig. 7). We also found that the resilience of the bacterial community to drought (calculated as pairwise Bray–Curtis similarities between control and

drought communities) was linked to the resilience of the plant community, but this relationship was not significant for fungal communities (Fig. 6g, h).

We further inferred the potential mechanisms through which plant community change affected fungal and bacterial community composition as well as functional guilds involved in denitrification and $N_2O$ reduction by constructing a structural equation model (SEM) for the final, late recovery sampling (Supplementary Fig. 4 and Supplementary Note 1). We found that the increase in *D. glomerata* biomass was directly associated with bacterial community composition. The increase in *D. glomerata* was also indirectly associated with fungal and bacterial community composition through reducing soil moisture content, which was caused by an increase in aboveground biomass and thus evapotranspiration (Fig. 8). The abundances of *nir* and *nosZ* genes, used as proxies for denitrifiers and nitrous oxide reducers, respectively, were not predicted by changes in plant community composition. Surprisingly, soil inorganic and organic-dissolvable nitrogen, and dissolved organic carbon were not associated with bacterial or fungal community composition, despite our observation that the *D. glomerata*-associated increase in aboveground biomass reduced soil inorganic N availability (regression; $R^2 = 0.09$, $P = 0.010$).

**Links with ecosystem functioning**. We constructed a SEM to assess the direct and indirect effects of drought on soil functioning (Supplementary Fig. 5 and Supplementary Note 2). We used a multigroup modelling approach to assess how our hypothesised relationships between plant community composition, microbial communities, and $CO_2$ and $N_2O$ fluxes changed during drought and recovery[9]. Drought strongly and directly affected

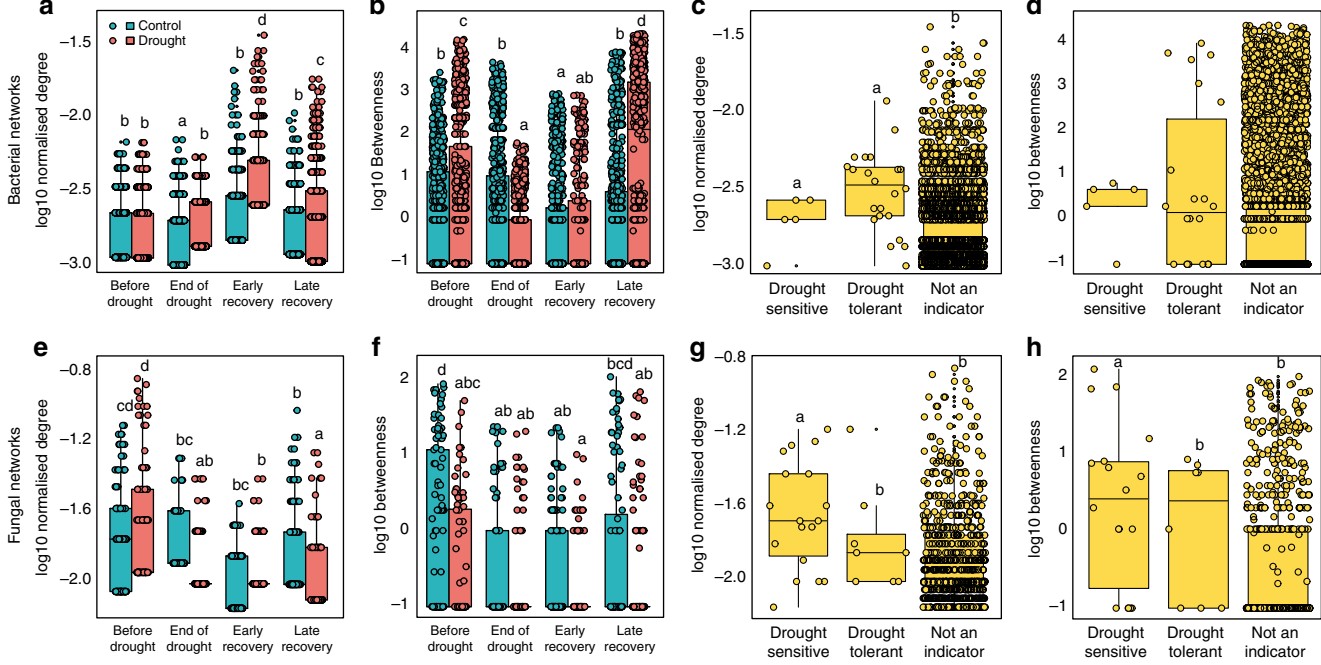

**Fig. 5** Node connectedness and centrality of bacterial and fungal networks. In bacterial networks, drought increased connectedness (normalised degree) and centrality (betweenness) at post-drought samplings (**a, b** Sampling × Drought interaction $F_{3,5973} = 67.3$, $P < 0.001$ and $F_{3,5973} = 107.5$, $P < 0.001$, respectively), while in fungal networks, drought did either not affect, or decrease these properties (**e, f** Sampling × Drought interaction $F_{3,833} = 21.6$, $P < 0.001$ and $F_{3,833} = 1.4$, $P = 0.234$ for normalised degree and betweenness, respectively). In bacterial networks, the normalised degree of drought-sensitive and drought-tolerant indicator OTUs was higher than those of non-indicator OTUs (**c** $F_{2,5977} = 12.5$, $P < 0.001$), but betweenness was not affected (**d**). In fungal networks, drought-sensitive OTUs had a higher normalised degree and betweenness than non-indicator OTUs (**g, h** $F_{2,834} = 10.8$, $P < 0.001$ and $F_{2,834} = 11.6$, $P < 0.001$, respectively). Lines in boxes represent median, top and bottom of boxes represent first and third quartiles, and whiskers represent 1.5 interquartile range; dots represent single observations. Boxes with the same lower-case letters are not statistically different

both fungal and bacterial community composition at the end of the drought through changing soil moisture, but plant community composition only affected bacterial community composition directly (Fig. 9). Neither fungal nor bacterial community composition were linked to ecosystem respiration, but bacterial community composition was linked to the abundance of *nir* genes. However, 1 week after rewetting, bacterial but not fungal community composition was directly associated with both ecosystem respiration and $N_2O$ production, and indirectly with $N_2O$ production through its link with the abundance of *nosZ* genes. As expected, the abundance of *nir* genes was linked to increased $N_2O$ production, while *nosZ* gene abundance linked to reduced $N_2O$ production at early recovery. At the final sampling, 2 months after ending the drought, this association of *nir* and *nosZ* gene abundances ratio with $N_2O$ fluxes disappeared (Fig. 9), and all rates of ecosystem processes had returned to control levels (Supplementary Fig. 6) irrespective of drought, *nir* and *nosZ* gene abundances increased during the duration of the experiment (Supplementary Fig 7). The $N_2O$ reducers were dominated by *nosZ* clade I indicating that complete denitrifiers rather than non-denitrifying $N_2O$ reducers were promoted in the experiment[27]. We did not find any relationship between bacterial and archaeal ammonia oxidiser gene abundances and $N_2O$ emissions (see Supplementary Fig. 8 for drought effects on *amoA* gene abundances).

## Discussion

Our findings provide insight into the immediate and delayed response of belowground networks to drought and highlight the role of plant community composition in governing the dynamics of this response. We show that soil bacterial and fungal networks have different properties and respond differently to drought, with

drought having a much stronger impact on bacterial than on fungal networks. We also found that drought-induced changes in plant communities had long-lasting associations with bacterial networks and communities and strongly governed their recovery, but much less so for fungal networks and communities. Bacterial co-occurrence networks were characterised by properties that indicate low stability under disturbance, such as high connectivity and centrality, and low modularity, while fungal networks had properties that suggest higher stability. In bacterial networks, abundant, drought responsive indicator OTUs were highly central and connected, regardless of whether they were drought-sensitive or drought-tolerant (Fig. 5), suggesting that they might drive the observed drought-induced changes in bacterial networks[26,28,29]. The central indicators of drought in the bacterial networks, *Rhodoplanes* and DA101, are highly abundant in soils in general, although little is known about their ecology[30]. However, their drought tolerance might be a result of their ability to maintain dominance under a wide range of environmental conditions.

The most dominant bacterial taxa were the strongest responders to drought (negative correlation between abundance and $P$ value of drought response), which is in contrast to previous studies[31,32], and we found that these dominant taxa drive network structure (positive correlations between connectedness and betweenness, and abundance). Thus, the most abundant OTUs were driving bacterial, but not fungal, network reorganisation in response to drought[26,28,29]. While the relationships between bacterial phylogeny and function are complex[33,34], shifts in the abundance of indicator taxa might inform on the stability of these networks, and consequently on the response of soil bacterial communities to drought.

Drought caused a long-lasting shift in plant community composition, and this shift was strongly associated with bacterial

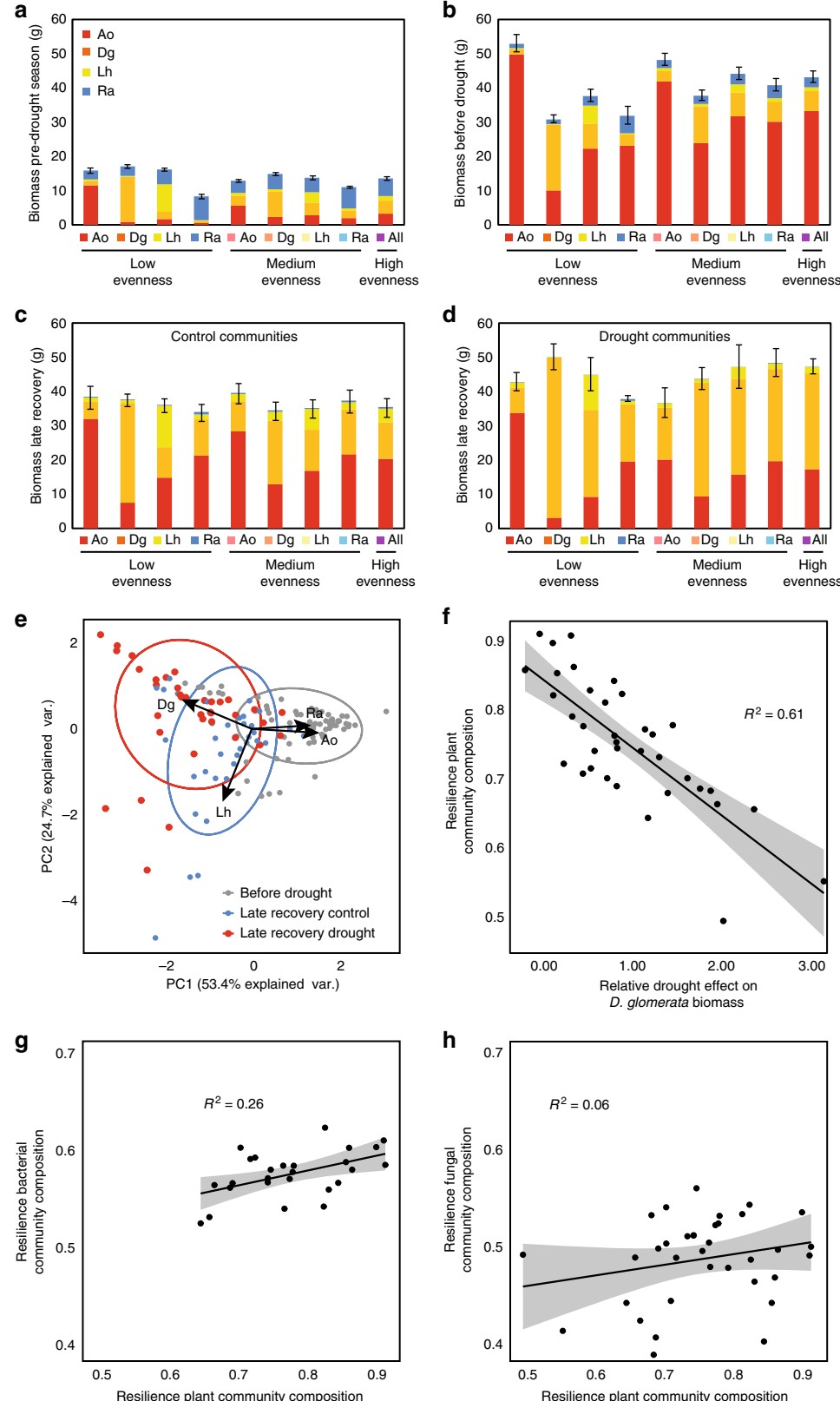

communities and networks. Fungal communities have been found to be more responsive than bacterial communities to vegetation change[35], and there is accumulating evidence that fungi are the first consumers of belowground inputs of plant-derived C[24,36]. However, our findings point to a novel mechanism

by which vegetation change can affect soil microbial community reorganisation via changes in soil moisture content. Drought increased the abundance of the fast-growing grass *D. glomerata*[37], resulting in increased total aboveground biomass, which in turn resulted in a prolonged reduction in soil moisture due to higher

**Fig. 6** Drought effects on plant community composition and the relationship between changes in plant community composition and microbial community composition. Biomass of the four species in our established plant communities in the pre-drought season (**a**), just before drought (**b**) and 2 months after ending the drought (**c**, **d**), split for control (**c**) and drought communities (**d**). Plant community treatments are on the x-axis, with three different evenness levels and each species dominating within each evenness level. Total community biomass increased marginally in response to drought (ANOVA $F_{1,48}$ = 273, $P$ = 0.08); *D. glomerata* (Dg) biomass increased strongly under drought, except in *Anthoxanthum odoratum* (Ao)-dominated communities (ANOVA dominant species × Drought interaction $F_{14,48}$ = 4.52, $P$ = 0.004). Ao *Anthoxanthum odoratum*, Dg *Dactylis glomerata*, Lh *Leontodon hispidus*, Ra *Rumex acetosa*. PCA-biplot of plant community composition (**e**) shows that PC-axes 1 and 2 scores were significantly affected by drought at the late recovery sampling (ANOVA $F_{1,48}$ = 273, $P$ < 0.001 and $F_{1,48}$ = 37.2, $P$ = 0.008, respectively). Resilience of plant community composition (similarity between drought and control, measured as Bray–Curtis distances) explained by the relative change in *D. glomerata* biomass in response to drought (**f**). With a larger drought-induced increase in *D. glomerata* biomass, droughted plant communities were less similar to control communities ($P$ < 0.001). The resilience of bacterial community composition (**g**) was positively explained by the resilience of plant community composition ($R^2$ = 0.26, $P$ = 0.008), but this relationship was not significant for fungal community resilience (**h** $R^2$ = 0.06, $P$ = 0.182). In **a–d**, bars represent means ± 1 SE ($n$ = 4, SE for total biomass only); in **f–h**, dots represent single observations, with shaded areas indicating 95% confidence intervals

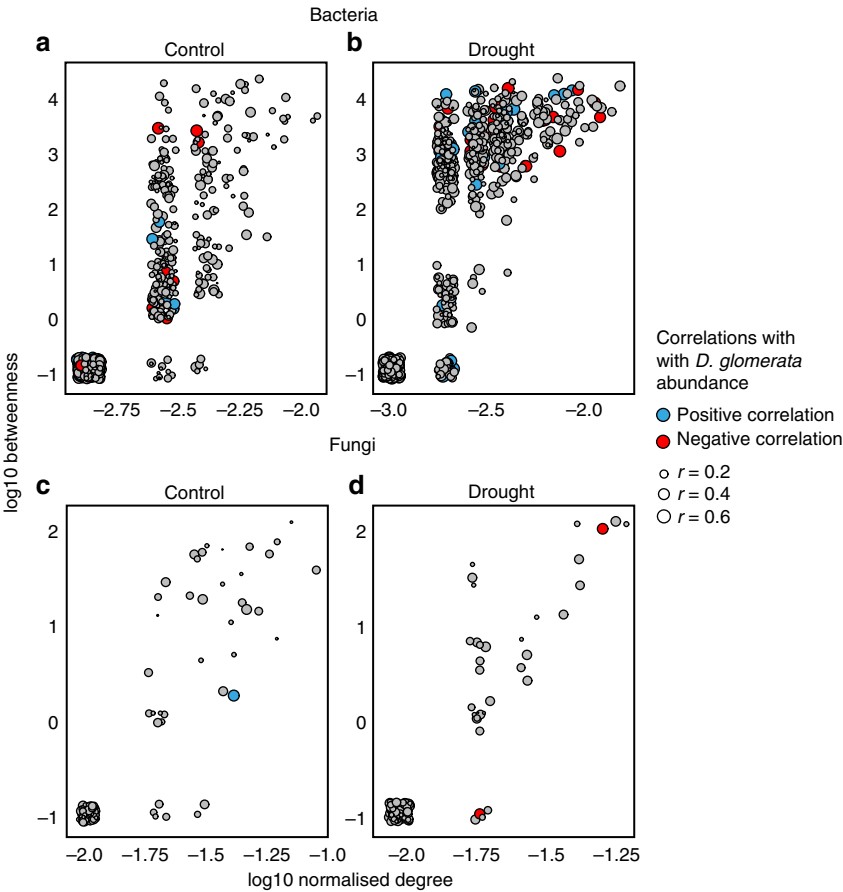

**Fig. 7** Relationship between network properties and correlation with *D. glomerata* biomass at the late recovery sampling time point. Symbol size indicates the strength of the correlation of that node with *D. glomerata* abundance; blue and red symbols indicate significant positive and negative correlations with *D. glomerata*, respectively, while grey symbols represent non-significant correlations. All networks had a positive correlation between node-normalised degree and betweenness (**a–d**), but only in bacterial drought networks, there was a strong positive relationship between both normalised degree and betweenness and the strength of node correlation with *D. glomerata* abundance (ANCOVA normalised degree × Drought interaction $F_{1,1676}$ = 25.9, $P$ < 0.001 and Betweenness × Drought interaction $F_{1,1676}$ = 14.5, $P$ < 0.001 for bacteria (**b**); and Normalised degree $F_{1,225}$ = 1.1, $P$ = 0.742, Normalised degree × Drought $F_{1,225}$ = 1.2, $P$ = 0.275, Betweenness $F_{1,225}$ = 0.08, $P$ = 0.772, Betweenness × Drought $F_{1,225}$ = 0.678, $P$ = 0.411 for fungi (**d**))

evapotranspiration. This increase in *D. glomerata* is likely explained by the tolerance of this species to drought and its ability to capitalise on the flush of N that occurred on rewetting, thereby enforcing its dominance relative to other species[38]. Our SEM (Fig. 8) showed that while plant community induced changes in soil moisture affected both bacterial and fungal community composition 2 months after the drought had ended, only bacterial community composition was directly affected by the biomass of

*D. glomerata*. This corresponds with our observation that *D. glomerata* biomass was linked to central and connected OTUs in bacterial networks, but not in fungal networks. In addition, both the immediate and delayed reductions in moisture affected bacterial communities more than they did fungal communities. Thus, drought can have long-lasting legacy effects on soil microbial communities by promoting the dominance of a fast-growing grass. Such responses could potentially be common given

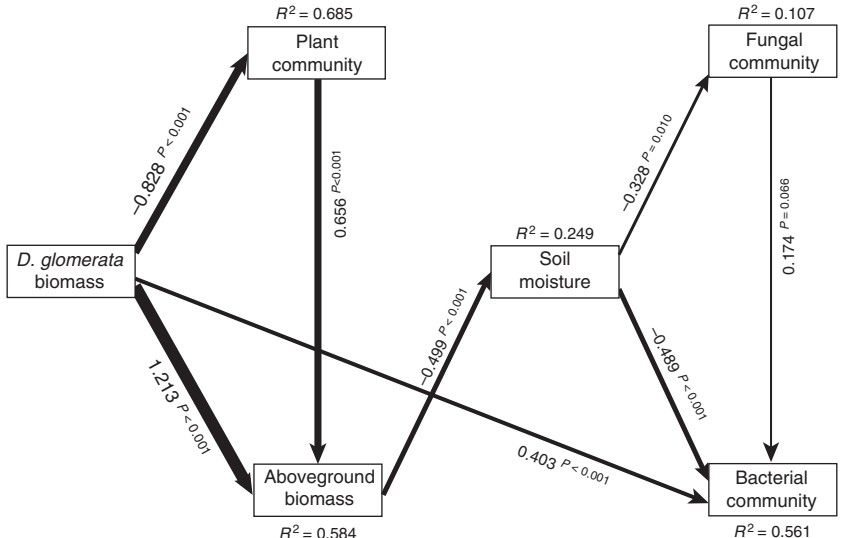

**Fig. 8** Structural equation model of relationships between *D. glomerata* biomass, plant community attributes and microbial properties, at the final, late recovery sampling. For each significant relationship, standardised coefficient and *P* values are given alongside arrow, and arrow weights are proportional to standardised coefficients. The model fit the data well: $P = 0.345$, CFI $= 0.995$, $P$ RMSEA $< = 0.05 = 0.439$

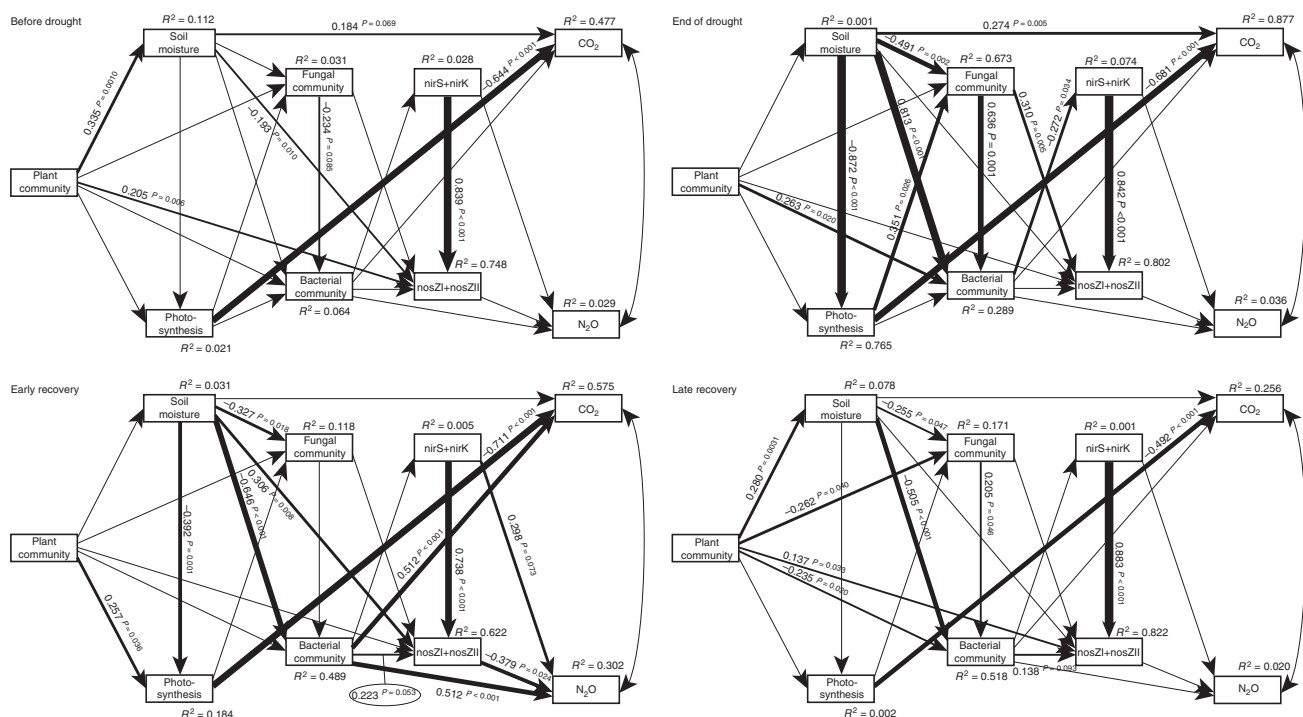

**Fig. 9** Structural equation model of relationships between plant community attributes, microbial properties and ecosystem functioning for each sampling. For each significant relationship, standardised coefficient and $P =$ values are given alongside arrow, and arrow weights are proportional to standardised coefficients. Our multigroup model fit the data well: $P = 0.309$, CFI $= 0.994$, $P$ RMSEA $< = 0.05 = 0.574$

that increases in abundance of fast-growing grasses are often reported following drought, contributing to the maintenance of grassland productivity[38–40]. While these vegetation-mediated legacy effects on soil microbial communities did not result in long-term-altered soil functioning, they were shown in a related study to feedback positively on growth of *D. glomerata*[5], thereby potentially reinforcing its dominance. As such, changes in soil microbial communities resulting from drought have the potential to have legacy effects on plant community composition, in this case triggering a positive feedback on *D. glomerata* abundance[5],

potentially contributing to long-term-altered plant community composition.

Not only were bacterial communities affected more by drought, both directly and indirectly through vegetation change, but bacterial communities also showed stronger links to $CO_2$ and $N_2O$ fluxes than did fungal communities during drought recovery. As expected, bacterial communities predicted *nosZ* gene abundances[27], and both $N_2O$ emissions and abundances of genes indicative of denitrifier genes (*nir* and *nosZ* clade I) were sharply increased 1 week after rewetting. These dynamics, together with

the low variance of $N_2O$ explained in our SEM, suggest that the flush in inorganic N availability after rewetting (which is not included in the SEM) caused growth of denitrifiers[41] that did not use the final step in the pathway under N replete conditions. Thereby, $N_2O$ emission rates increased, but once available N content decreased, the denitrifiers likely used the entire pathway and terminated with nitrogen gas. This would explain why gene abundances remained high at the final sampling, while $N_2O$ production returned to control levels. Our observations are in contrast with previous studies that show a reduction or no response of denitrifier genes in response to drought[42,43]. Moreover, we show that the indirect effects of drought can increase both denitrifier and ammonia-oxidiser gene abundances, potentially 'priming' N cycling communities to respond to increased N availability after a second drought.

Our findings have important implications for understanding how complex soil microbial communities respond to climate extremes and other disturbances. While the exact nature of inferred bacterial and fungal interactions remains unknown, our data show that drought promoted destabilising properties of bacterial co-occurrence networks. We also reveal an important association between drought-induced shifts in plant community composition and changes in bacterial communities, bacterial networks and the relative abundance of denitrification and nitrous oxide reduction genes long after the drought has ended. These findings suggest that bacterial communities might not be as resilient as previously thought, and that changes in vegetation resulting from drought could have long-lasting effects on soil bacterial communities, potentially influencing plant community composition and the ability of aboveground and belowground communities to withstand future disturbances[5,6,9,44].

## Methods

**Experimental set-up and soil and greenhouse gas analyses**. Topsoil (silt loam of the Brickfield 2 association (Avis & Harrop, 1983), pH 6.2, C and N content 3.13 and 0.25 g $kg^{-1}$, respectively) was collected from mesotrophic grassland at Hazelrigg Field Station, Lancaster University, UK (54°1′N, 2°46′W, 94 m a.s.l), where the experiment was conducted. Average yearly rainfall at the site is 1092 mm, average yearly temperature is 9.3 °C, based on daily site measurements since 1966. After homogenisation and removal of stones and roots, 72 pots of 42L (38 × 38 cm, 40 cm depth) were filled with soil in May 2012 and planted nine different plant communities consisting of 36 individuals, arranged in a randomised block design. Each community consisted of four plant species (*Anthoxanthum odoratum*, *Dactylis glomerata*, *Leontodon hispidus* and *Rumex acetosa*, germinated from seed in glasshouse conditions) and plant communities differed in evenness (low, medium and high) and dominant plant species (each species alternating in the low and medium evenness treatment; the high evenness treatment had equal abundances of all species; see Supplementary Table 2). Plant communities were left for two growing seasons, and during the second growing season, an extreme drought designed to simulate a 100-year drought event was imposed in a split-block design by placing transparent rain covers over the pots from May to July 2013, following Bloor and Bardgett[45]. All plant community treatment and drought combinations were replicated four times. Local weather data (1967–2008) were used to fit a Gumbel I distribution to the annual extremes of drought duration for the local growing period. During the experimental drought, 75.6 mm of rainfall was excluded in June, 84.9 in July and 131.1 mm in August (see Supplementary Fig. 12). All pots, and control treatments during the drought, were supplemented with 700 ml of additional water (equivalent to 0.5 mm of rainfall) every 2 days in dry weeks. Aboveground biomass was cut to 4 cm two times between plant community establishment and start of the drought to simulate grazing (end of growing season 2012 and 1 week before starting the drought in 2013), and the biomass of the four individual species was dried (70 °C) and weighed for each pot. Samples were taken to quantify microbial properties and gaseous fluxes immediately before starting the drought, at the end of the drought, and 1 week and 2 months after ending the drought. At each sampling date, five randomly distributed soil samples per pot were taken (1 cm diameter, 0–10 cm depth), pooled, sieved (2 mm) and kept at −80 °C until processing. $CO_2$ fluxes (ecosystem respiration and net ecosystem exchange) were measured using a portable Infra Red Gas Analyser (EGM4, PP Systems) using a blacked-out and transparent chamber (to prevent and allow photosynthesis), respectively. Photosynthesis rates were calculated as net ecosystem exchange minus respiration rates, with negative values representing an uptake of $CO_2$. $N_2O$ fluxes were measured using a static chamber approach as in Ward

et al.[46]. Briefly, dark, airtight chambers fitted with a septum for gas collection were placed over each mesocosm, and headspace was sampled manually every 10 min for 30 min (four time points) using a syringe (10 ml) fitted with a needle. Samples were injected into 3 ml vacuum exetainers (Labco, Lampeter, UK) and analysed by gas chromatography using Autosystem XL GCs (Perkin Elmer, Waltham, MA, USA). Fluxes were adjusted for field temperature when sampling, headspace volume and chamber area, and calculated by linear regression using all time points sampled. At the final sampling (September 2013), a full sampling of plant and soil properties was done. Aboveground vegetation was cut to 4 cm, split per species, dried (70 °C) and weighed. Three soil cores (3 cm diam, 10 cm depth) were pooled per pot, homogenised, sieved (4 mm) and used for analysis of water soluble soil carbon (dissolved organic C—DOC) and nitrogen ($NO_3^-$, $NH_4^+$, dissolved organic N—DON) pools. For dissolved organic C and N, 5 g of soil was extracted using 35 ml of Milli-Q water, extracts were filtered (0.45 μm), and analysed using a 5000A TOC analyser (Shimadzu, Japan) (DOC) and a AA3 HR Auto Analyzer (Seal Analytical, UK) (DON). For inorganic N, 5 g of soil was extracted using 25 ml of 1 M KCl, filtered through Whatman no 1, and analysed on a AA3 HR Auto Analyzer (Seal Analytical, UK). One core (3 cm diam, full depth of pot) was used for determination of total root biomass and root trait analysis. Structural root traits were analysed using WinRhizo® root analysis software (Regent Instruments Inc., Canada) and an Epson flatbed scanner. After analysis, roots were blotted dry, weighed, dried at 70 °C and re-weighed to calculate root dry matter content (RDMC). Specific root length (SRL) was calculated by dividing the dry biomass by the total root length (cm $g^{-1}$); root tissue density (RTD) was calculated by dividing the weight of the dry biomass by the root volume (g $cm^{-3}$).

**Quantitative PCR of *nir* and *nosZ* genes**. DNA was extracted from 0.3 g of soil using the MoBIO PowerSoil-htp 96-Well DNA Isolation kit (Carlsbad, CA) according to the manufacturer's protocols. The DNA quality was checked by agarose gel electrophoresis and prior to performing quantitative PCR, the quantity was measured using a Qubit fluorimeter (Invitrogen, Carlsbad, CA). To determine the genetic potential for denitrification and $N_2O$ reduction, the genes *nirS* and *nirK*, coding for the cytochrome-like nitrite reductase and the copper-dependent nitrite reductase in denitrifiers, and *nosZI* and *nosZII*, coding for the nitrous oxide reductase from clade I and II, were quantified by quantitative PCR based on SYBR green detection using a Biorad CFX Connect Real-Time System (Biorad) and according to the amplification protocols in Supplementary Table 3. Each reaction had a volume of 15 μl using iQ™ SYBR® Green Supermix (Bio-Rad Laboratories, Inc.), 0.05% bovine serum albumin, 0.5 μM of each primer except 0.25 μM for the *nirK* primers, and 5 ng DNA. Standard curves for each gene were obtained by serial dilutions of linearised plasmids containing fragments of the respective genes, which were obtained from pure cultures. Standard curves were linear ($R^2 = 0.990$) in the range used. The amplifications were verified by melting curve analyses and agarose gel electrophoreses, and non-template controls resulted in negligible values. Prior to quantification, potential inhibition of the PCR reactions was checked for each sample by amplifying a known amount of the pGEM-T plasmid (Promega) with the plasmid-specific T7 and SP6 primers when added to the DNA extracts or non-template controls. No inhibition of the amplification reactions was detected with the amount of DNA used.

**Quantitative PCR of archaeal and bacterial *amoA* genes**. Archaeal and bacterial *amoA* gene abundances were determined by real-time PCR in a RealPlex2 Mastercycler (Eppendorf, Stevenage, UK) using QuantiFast® SYBR® Green PCR Master Mix (Qiagen, Crawley, UK), as described by Thion and Prosser[47]. Briefly, 20 μl final volume reaction comprised of 0.2 mg $ml^{-1}$ bovine serum albumin (BSA), 1.5 μM of both primers, 10 μl of QuantiFastTM qPCR master mix (Qiagen, Crawley, UK) and 2 μl DNA template, diluted to reach 5–10 ng $μl^{-1}$. CrenamoA23f/CrenamoA616r[48] and amoA-1F/amoA-2R[49] primers were used to amplify AOA and AOB *amoA* genes, respectively (Supplementary Table 3). Two complete standard dilution series, prepared as described by Thion and Prosser[47], from $10^1$ to $10^7$ AOA or AOB *amoA* gene copies, were run for every real-time PCR reaction. Amplification efficiency ranged from 83% to 94% and from 92% to 101% for AOB and AOA, respectively, and only reactions with $r^2$ values ≥0.98 were considered.

**16S and ITS amplicon sequencing**. Amplicon libraries were constructed according to the dual indexing strategy of Kozich et al.[50], with each primer consisting of the appropriate Illumina adaptor, 8-nt index sequence, a 10-nt pad sequence, a 2-nt linker and the gene-specific primer. For bacteria, the V3–V4 hypervariable regions of the 16S rRNA gene were targeted using primers 341F[51] and 806R[52]; for fungi, the ITS2 region was amplified using fITS7f and ITS4r primer sequences described by Ihrmark et al.[53]. Amplicons were generated using a high-fidelity DNA polymerase (Q5 Taq, New England Biolabs) and pooled. PCR was conducted on 20 ng of template DNA employing an initial denaturation of 30 s at 95 °C, followed by (25 for 16S and 30 cycles for ITS) 30 s at 95 °C, 30 s at 52 °C and 2 min at 72 °C. A final extension of 10 min at 72 °C was also included to complete the reaction.

Amplicon sizes were determined using an Agilent 2200 TapeStation system (~550 bp: 16S; ~350–425: ITS) and libraries normalised using SequalPrep Normalisation Plate Kit (Thermo Fisher Scientific). Library concentration was calculated using a SYBR green quantitative PCR (qPCR) assay with primers specific

to the Illumina adaptors (Kappa, Anachem). Libraries were sequenced at a concentration of 5.4 pM with a 0.6 pM addition of an Illumina-generated PhiX control library. Sequencing runs, generating 2 × 300 bp reads, were performed on an Illumina MiSeq using V3 chemistry. The read 1 (R1), read 2 (R2) and index sequencing primers used were also 16S specific: R1 = sequence of the combined pad, linker and 341F; R2 = sequence of the combined pad, linker and 806R; I = reverse compliment of the R2 primer.

**Bioinformatic and statistical analyses of sequencing data**. Sequenced paired-end reads were joined using PEAR[54], quality filtered using FASTX tools (han-nonlab.cshl.edu), length filtered with the minimum length of 300 bpsm, presence of PhiX and adaptors were checked and removed with BBTools (jgi.doe.gov/data-and-tools/bbtools/), and chimeras were identified and removed with VSEARCH_-UCHIME_REF[55] using Greengenes Release 13_5 (at 97%)[56]. Singletons were removed and the resulting sequences were clustered into operational taxonomic units (OTUs) with VSEARCH_CLUSTER[55] at 97% sequence identity[57]. Representative sequences for each OTU were taxonomically assigned by RDP Classifier with the bootstrap threshold of 0.8 or greater[58] using the Greengenes Release 13_5 (full)[56] as the reference. Unless stated otherwise, default parameters were used for the steps listed. The fungal ITS sequences were analysed using PIPITS[59] with default parameters as outlined in the citation. Briefly, this involved quality filtering and 97% clustering of the ITS2 region as indicated above for the 16S processing, using the UNITE database for chimera removal and taxonomic identification of representative OTUs. Subsequent analyses of taxon abundances were conducted in R using principally the vegan library (https://github.com/vegandevs/vegan) for rarefaction and ordination. Both bacterial and fungal OTU abundance tables were resampled to a minimum of 4000 reads per sample, and samples with a Shannon diversity higher than 6 and 3.2 for 16S and ITS, respectively, were removed prior to further analyses.

We generated non-metric multidimensional scaling (NMDS) plots to visualise and determine the effect of drought on fungal and bacterial community composition using the functions decostand and adonis in the R library vegan. We also performed principal components analysis (PCOA) based on Bray–Curtis dissimilarities on both communities using the function cmdscale; PCOA scores were used as proxies for community composition in subsequent structural equation modelling. In addition, we used Bray–Curtis similarities between control and drought fungal and bacterial communities as a measure of community resistance and resilience to drought. To identify OTUs significantly associated with the drought treatments at each time point, we used indicator species analysis as implemented within the R library labdsv (http://ecology.msu.montana.edu/labdsv/R). The indval score for each gene is the product of the relative frequency and relative average abundance within each treatment, and significance was calculated through random reassignment of groups (1000 permutations). In the circle plots and subsequent network analyses we focus on only the abundant indicator OTUs which are significant (P < 0.05) and present at >1% abundance. Full tables of indicator scores are provided in supplementary material. For calculating indicator taxa and generating circle plots all taxa with fewer than ten reads across all samples were also excluded to ease presentation.

We analysed fungal and bacterial networks for each sampling and drought and control treatment separately. Thus, each network was based on 36 communities, but only OTUs that occurred in at least 8 communities were included in the analysis, as in Shi et al.[28]. All network analyses were done in R, using the package igraph, as in Williams et al.[26]; we adapted the code available at https://github.com/ryanjw/co-occurrence. Interactions consisted of Spearman's rank correlations and co-occurrence networks were constructed using only significant correlations (P < 0.01 as in Barberan et al.[21]) of ρ > 0.6; this cutoff was chosen to include a range of interactions strengths (not only strong interactions). Random networks were constructed as in Williams et al.[26]. Subsequently, we detected network modules using the edge.betweenness.community function and calculated clustering coefficients using the transitivity function, analysed the centrality of network nodes using the betweenness function, and analysed the connectedness of network nodes using the degree function. All bacterial and fungal networks were significantly more clustered than random networks. Networks were visualised in the R library igraph.

**General statistical analyses**. All data were checked for normality and log-transformed if necessary. Gas fluxes, N cycling gene data and measures derived from sequencing data were analysed using repeated measures ANOVA taking into account the split-plot design (error term with block, drought and mesocosm), soil nutrient data were analysed using ANOVA (error term with block and drought). All analyses were performed in R version 3.3.2[60].

**Structural equation modelling**. We constructed our a priori models based on current knowledge on plant-microbe-functioning interactions and tested whether the data fit these models using the standard and the multigroup modelling approach in the R library lavaan. We used model modification indices and stepwise removal of non-significant relationships as in De Vries and Bardgett[61]. We used a minimum set of parameters to assess model fit, including root mean square error of approximation (RMSEA), and comparative fit index (CFI). We used PCOA axis 1 scores as a proxy for fungal and bacterial community composition, PCA axis

1 scores as a proxy for plant community composition, and the ratio between the sum of *nirS* and *nirK* and the sum of *nosZ*I and *nosZ*II gene abundances to indicate relative changes in genetic potential for the soil functions denitrification and N₂O reduction.

**Code availability**. Code for network analysis was adapted from, and is available at, https://github.com/ryanjw/co-occurrence. Code for indicator analysis is available on request.

**Data availability**. OTU tables and metadata are available from figshare (https://doi.org/10.6084/m9.figshare.6548999.v1). Raw sequence data have been deposited at the EBI European Nucleotide Archive under study accessions ERP109485 (fungal amplicons) and ERP109472 (16S amplicons).

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

## Acknowledgements

This study was funded as part of the European project EcoFINDERS (FP7-264465). F.T. d.V. is supported by a BBSRC David Phillips Fellowship (BB/L02456X/1). We thank Caley Brown, Melanie Hartley, Neil Mullinger, Helen Quirk, Deborah Ashworth and Victor van Velzen for help in the lab and field.

## Author contributions

F.T.d.V. and R.D.B. conceived and designed the experiment, with input from R.I.G., M. B., P.L., N.O. and J.I.P.; R.D.B. acquired the funding needed to initiate the study as part of the EcoFINDERS project led by P.L. F.T.d.V., R.I.G., H.C., M.G., H.S.G., S.H., A.K., A.M. K., M.K., E.L., K.E.M., A.O., C.T., B.T. and R.D.B. performed field sampling and laboratory analyses. F.T.d.V. and R.I.G. statistically analysed data, and F.T.d.V. led writing the manuscript in close consultation with R.D.B., R.I.G., S.H. and J.I.P., and with contributions from all authors.

## Additional information

**Competing interests:** The authors declare no competing interests.

