## [Peer Review File · Nature Communications]

Reviewers' comments:

Reviewer #1 (Remarks to the Author):

de Vries and colleagues report on an experiment where they examined the microbial community network response of model grasslands to drought. I found the design of the experiment to be very comprehensive and well thought-out, including different plant composition, different time points relative to the disturbance, and some assessment of function. Many types of analyses were utilized, and my concerns mostly revolved around the choice and justification of what was used when to address the main questions in the paper. With these concerns, I do not find these results and conclusions convincing and suggest further evidence to strengthen the conclusions.

How resistance (impact) and resilience are calculated relative to the experimental framework needs more explicit treatment. While these terms are used often, it was not clear what treatment comparisons were necessary to conclude something about resilience dynamics and differential resilience dynamics across groups. There are many possible comparisons needed to make these conclusions; the methods mention the use of B-C compositional dissimilarity, but it is not clear when that measure was used or the statistical comparison. As another example, the first paragraph of the paper states "Drought changes in bacterial communities link stronger to soil functioning than those in fungal communities" but it appears that that signal to function appears only in the recovery phase.

There were two network analysis conducted, correlation networks, which included both positive and negative interactions, and co-occurrence networks, which only consisted of positive correlations. This needs to be spelled out much more clearly. For instance, for correlation networks, a conclusion is that bacterial networks are more stable because of more negative interactions (In 95) and then, for co-occurrence networks, a conclusion is that bacteria co-occurrence networks (with just positive correlations) were larger and more connected, and so have lower stability than fungal networks (In 107).

Relatedly, the paper needs more justification for a couple of main methodological analysis decisions. It was not clear why just positive edges were included in the network co-occurrence analysis. Also, justification for the statistical analysis of the network statistics using each node within a community as a statistically independent point (e.g., Fig 2, 3) is needed. A hierarchical model that takes into account mesocosm-level replication or aggregation to the mesocosm-level may be appropriate.

The result starting In 137 confused me, as I was expecting a correlation result and instead several regression stats were listed. Also: what does this statement mean? "While these findings support recent findings that subordinate bacterial taxa respond strongest to precipitation changes, here we show that it is the abundant bacterial taxa that drive bacterial network structure."

It was interesting that the treatments started with the establishment of four types of plant communities, but the analyses consisted of correlations with *D. glomerata* abundance. Perhaps the authors could explain the original design and their reasoning here as bit more.

Lastly, I'd urge the authors to really think about their main conclusions from this paper concerning network stability given the overall finding that there were only weak links between microbial compositional changes and the functions measured. Rather than thinking about resilience in terms of composition, is it that changes in composition lead to resilience in function? One definition of resilience is the ability to reorganize to stabilize function.... Particularly since one of the main conclusions (last sentence in the first paragraph) is the importance to soil function. And were there no relationships between function and network structure?

Comments on writing:

Many very long paragraphs that lose the focus on their topic sentences

Ln 33-37 long sentence, lots of commas.

Ln 60-64, very long sentence.

Ln 79: is recovered the right word here?

Ln 85, comma in a strange place

Ln 105-112 is one long sentence, and tense changes.

Reviewer #2 (Remarks to the Author):

The paper by de Vries and colleagues examines the effect of intense drought on fungal and bacterial communities. In particular they build upon previous work in this area examining network properties. They find bacterial networks are less stable than fungal networks, which likely has implications for ecosystem function. Further, using SEM they link the functions of N₂O flux and denitrification genes to plants. Overall this is an interesting study with some issues that need to be addressed.

General comments

Overall this is a very interesting and well-written paper. It seems that this came from Nature and given that, the lack of detail is understandable. I, and presumably readers, would like to see expanded introduction and discussion that better sets the scene for the data.

My general issue with the study is the delineation between effects of drought and the indirect effects of drought mediated through the plants. The authors sample to 10cm and presumably sample the rhizosphere of these plants and if these grasses are like most grasses, there should be a lot of root biomass (the authors mention measuring it but don't report it). We see from the data that the bacterial networks are more sensitive than the fungi but there's no way to really say if this was simply due to drought or is an effect of rhizodeposition. Based on extended figure 7e there's a significant effect of drought/recovery on plant community composition and presumably the interactions between plants and microbes. Further, the authors measure geochemical parameters but don't report them anywhere and it would be helpful for determining if the effects of drought on the chemistry, which would be affecting the bacteria. The SEM might be able to get at this idea, but the authors need to tread carefully when discussing this and attribute the changes to both plants and the drought.

Throughout the authors could use more explanation of their data and there's certain places they could dig deeper in their interesting results. Further, there are several places where there's not enough information to figure out how the authors did what they did (i.e. CH₄ fluxes, NEE calculations, etc).

Abstract

Line 30: the authors don't address negative interactions in this paper. But they probably should.

Line 34: I would mention that they are mesocosms, not grasslands.

Introduction

The introduction, as a whole, needs more detail as it only stands as 1 true paragraph that's bridged between the abstract and the rest of the main text.

Lines 60-64: text like this is very helpful for interpreting the network metrics as what they mean can be difficult to remember.

Line 76: N₂O flux is the result of not just denitrification, but also ammonia oxidation and fungal denitrification.

Results/Discussion

Line 77: it would be nice to see the R-values for the PERMANOVA. This would be informative to interpreting the stats in the figures. Also, was drought the only parameter used in the PERMANOVA?

Line 78: an increase of 0.07 in fungal evenness isn't a dramatic change in community evenness, same can be said about the decrease in bacterial evenness.

Line 84: It is important to note that fungi sporulate as well. Further, bacterial dormancy would explain a lack of change in community composition, but the authors do see a change for bacteria at all times following drought stress. In addition, I do not buy the argument about osmotic stress killing off bacteria since there's considerable osmotic stress from the drought itself.

Line 100: why only positive correlations? I didn't see justification for this and would seem to be an interesting component for this study to identify negatively responding taxa and how they're network metrics respond.

Line 113: very cool result!

Lines 118-120: this is cool. It also makes me want to know who is responding here.

Lines 128-129: these r-values are very, very low.

Line 166-167: these results makes me question the main interpretation of the data. Are the plants driving this or is it the drought?

Line 176: The authors, somewhere, need to address why they only targeted denitrification when looking at N₂O flux and not ammonia oxidation. We know that ammonia oxidation can account for a substantial portion of N₂O flux (i.e. Santoro et al. 2011, Science). Further, since fungi were targeted in this study, the authors should discuss fungal denitrification as its only produces N₂O (e.g. Maeda et al. 2015, Scientific Reports).

Line 179: this specific plant is only part of the plant biomass in the SEM, right? Also the biomass is aboveground?

Lines 181-182: wouldn't this primarily be due to evaporation (direct and from the plant)?

Line 182-183: was bacterial-N interactions examined? Seems like bacteria should have some correlation.

Lines 184-188: I think this result is particularly exciting and I think it need to come out more here (with discussion of priming and legacy effects) and in the abstract.

Line 190: would be great to discuss the potential feedback on ecosystem function.

Lines 188-190: this gets at my primary concern with the paper. The direct effects of drought are likely not discernable from the effects of plants here.

Lines 197-198: this seems odd as we'd expect they would. Perhaps diversity would better link this?

Line 202-205: it does suggest redundancy only if the function remained the same as in control levels. Since no data is presented, I cannot determine if that's the case.

Line 208: can you better link these network properties to the fluxes/denitrification genes?

Lines 212-214: this could also be due to legacy effects in the soil.

Methods

Given that this is a drought study it would be nice to know what the temperature profile in the area as well as how much water the control plants received. Further, there's no mention of how photosynthesis was measured (which is in the SEM)?

While not a requirement, but it would be interesting to get at fungal function with something like FunGuild and see how that changes with drought. Since the denitrification genes (typically, since I don't know what primers were used) quantified are bacterial and don't target fungal nirK.

Line 222-224: Did the plants receive any additional watering or was it just reliant on rain?

Lines 224-226: I know little of the plants in the area of the world, but are these common plants? It would be helpful to know the rationale as to why the plants were selected.

Line 231: are the rain covers just little roofs? Or are they boxes? Also, do these effect the temperature in the area?

Lines 235-236: is there a reason that grazing was simulated three times throughout the experiment? I

know grazing is a stress plants have to deal with, however since the authors are looking at soil (likely including rhizosphere) the changes in root exudates can themselves effect composition of microbes. It would be helpful to know, in relation to the sampling dates, when the plants were clipped.

Lines 239-241: were the samples taken to be representative of all the plants in the pot? Were there roots in the samples? If so were they removed? The roots aspect of my question is important due to the effects of rhizodeposits on the communities.

Lines 253-256: please list the primers used. It is hard to evaluate how encapsulating the qPCR results are of denitrifying organisms. Further, what was used for standards? How was the DNA quantified? Was the DNA normalized for each reaction?

Line 274: 18S?

Lines 282-283: where was sequencing performed?

Line 287: since the authors are comparing fungal and bacterial networks it would be nice to have the sequences processed in the same manner and have that better delineated in the methods.

Lines 296-298: I have to question the efficacy of using a de novo cluster for fungal ITS reads. Given the range of sequence length (~75bp according to the bioanalyzer data in the text) the diversity estimates (both beta and alpha) would likely be affected, as well as the networks. While it shouldn't change the results dramatically, I would recommend doing a closed reference pick against UNITE to assuage the worry that OTUs are being improperly called because of the differences in sequence length, instead of biology.

Line 300: 'OTU' not out. Also, while I don't agree with the discussion on rarefaction, there's issues with rarefaction (see papers by McMurdie & Holmes). Have the authors tried raw abundances for their data? Or robust rarefaction (i.e. 10,000 resamples)?

Line 323: what methodology was used to determine significance and what alpha?

Line 323-324: seems like this sentence is missing something at the end. Please revise.

Line 330: it wasn't mentioned in the text at all or apparent in the SEM figure, so I may be wrong here and if I'm wrong, the authors need to expand on the SEM methods. But why do the SEM only show plants influencing the environment and not the other way around? Further, why aren't the C/reactive N included in the SEM? Both of these are likely to be affecting plants.

Further, since the authors make mention of plants responding to drought, the authors should consider including plant metrics (species, biomass) in the SEM. They can include plant species, perhaps as who's the dominant member at a given time, as a metric in the model as done recently in Nature Communications by Bowen et al. (2017).

Finally, why did the authors use the ratio of nirS/nirK and nosZ1/nosZ2? What do these ratios truly tell you about the system? It seems that the ratio just indicates the balance between the different types of denitrifiers (which doesn't correlate to N₂O emission potential) as well as the potential for N₂O reduction between canonical denitrifiers and N₂O scavengers. I would suggest that the authors adopt an approach akin to that of Jones et al., (2013), Nature Climate Change. These authors linked N₂O flux to the 4 genes the authors here quantified and that would be better suited to their SEM. The other thing that the SEM doesn't seem to take into account is the link between gas flux (in particular CO₂ which the plants will be making plenty of) and plant composition. This needs to be addressed somewhere in the model or text.

One last thing, wouldn't bacterial/fungal richness be more appropriate here in lieu of PCoA scores?

References

There are some inconsistencies in the references that should be resolved.

Data availability

Have the authors deposited the data and the associated metadata in a place like MG-RAST or the NCBI SRA? And while not mandatory by any means, I would enthusiastically suggest the authors make their

code available on GitHub (or equivalent site) so that others can follow the workflow, in particular the network analyses.

Figures

In general, the text of the figures are far too small and could use a size increase. Further, I don't like connecting the dots in the dot plot figures. It implies that there's no dynamic in between and I don't think we can make that assumption here.

Figure 1: These networks are tiny and are hard to make out the individual nodes. I'm not sure if anything can be done, but just an observation. It would also be nice to note the significance alpha value for the networks in the legend.

Figure 2: this is a personal preference, but the yellow-orange used in the boxplot is harsh. Also what do the boxes and lines mean? I assume quartiles and median but would be nice to know for sure.

Figure 3: how are you defining abundance of the plant? The number of shoots or leaves? What do the grey point mean, no correlation? Non-significant correlation?

Extended figure 1: the meaning of the grey bar in C-F isn't in the legend. I would also suggest perhaps using Shannon Diversity for C and D, overall richness isn't ideal for microbes.

Extended figure 2: maybe this was just my copy, but this figure was a mess and I couldn't not figure out what was going on except for in the first panel. Also, given how tiny these are is there a better way to represent this taxonomic data? Bar chart maybe?

Extended figure 7: no mention of how the beta diversity was calculated in the methods section. What does the grey shading refer to in g and h? Please put that information in the legend.

Extended figure 8: CH₄ fluxes are not mentioned anywhere in the text, please add to the text (results, methods) or remove panel e. I assume the bars are SE and dots are means? How was NEE calculated? No mention of that in the text.

Reviewer #3 (Remarks to the Author):

The authors present an interesting study on a highly relevant topic. They examine the effect of a climate change effect, extreme drought events, and test how plants communities as well as bacterial and fungal communities respond to drought and how they recover from it over the course of two years. They created an experimental set-up with plant communities that did or did not receive a drought treatment by rain shelters. The authors sampled prior to, during and two times after drought. From soil sample sequencing data they monitor the change in network structure of bacterial and fungal soil communities using co-occurrence networks and community composition (with NMDS) by looking at taxa richness, evenness, resilience, resistance, change, overlapping taxa, most responsive taxa to treatment with the use of Mantel tests, changes with similarity and dissimilarity indexes with the use of PCoA. Next to that they have measured respiration (CO₂-production) and N₂O-emissions from the system and related that to the gene expression of nirS/nirK and nosZ/nosZI to determine the genetic potential for denitrification and N₂O reduction and the nitrous oxide reductase with Q-PCR. Additionally, they performed a structural equation model to indicate cause and effect. They challenged several expectations. First, they expected bacterial networks to show more destabilizing properties, such as high connectance and low modularity, than fungi. Second, they also expected fungal communities to respond more in the recovery phase as changes in plant community structure and plant growth could be associated with the microbial recovery, but as fungi get the carbon first, they are expected to be most responsive. Third they expected drought-induced changes in fungal and bacterial networks to influence soil functioning. The authors confirmed hypothesis one with networks. My major concern with these networks is that also in the control networks there is also a large difference. And although it seems the networks are tested against random networks, I am wondering to what extent the differences in the networks both for the controls and the drought treatment is also

caused by seasonal effects as this experiment was monitored over 2 years. Drought stimulated *D. glomerata* in dominating into the mesocosms and that change seemed to influence the bacterial community most. This demonstrates hypothesis two is not supported. Although the authors performed a structural equation model to separate cause and effects, there seems to be some circularity in the reasoning. It is a bit of a chicken and egg discussion as soil communities influence plant communities already before drought is applied, then during and after drought plant community changes, but it is hard to determine if drought is causing a direct effect on plant growth, or changes the microbial community first which is then causing plant community shifts. Of course it could also be the other way around, but I do not get why the authors are so sure of that. Hypothesis three seems to be confirmed with a stronger effect from bacteria to gene expression. I did not see that immediately from the figures and I am just wondering if this is not slightly biased as I guess gene expression levels of *nirS/nirK* and *nosZ/nosZI* is more pronounced in bacteria over fungi, and then it is not surprising that they are more expressed in bacteria. But I might be wrong here, as I am not a specialist in the gene expression of these genes.

Minor points:

- Extended data figure 2 is extremely hard to read and understand. Is there a Mantel test behind this indicating the most responsive OTU's in the outer ring? There are also some corners missing in the figure.
- I am not sure how the community structure is one to one linked to gene-expression.

We thank all three reviewers for their thorough assessment of our manuscript and constructive feedback. We have now incorporated all suggested changes into our manuscript, and we have changed the manuscript formatting to the Nature Communications format, allowing us to include more text and figures. Below we provide a point-by-point response to all reviewers' comments (in italics).

Reviewer #1 (Remarks to the Author):

de Vries and colleagues report on an experiment where they examined the microbial community network response of model grasslands to drought. I found the design of the experiment to be very comprehensive and well thought-out, including different plant composition, different time points relative to the disturbance, and some assessment of function. Many types of analyses were utilized, and my concerns mostly revolved around the choice and justification of what was used when to address the main questions in the paper. With these concerns, I do not find these results and conclusions convincing and suggest further evidence to strengthen the conclusions.

How resistance (impact) and resilience are calculated relative to the experimental framework needs more explicit treatment. While these terms are used often, it was not clear what treatment comparisons were necessary to conclude something about resilience dynamics and differential resilience dynamics across groups. There are many possible comparisons needed to make these conclusions; the methods mention the use of B-C compositional dissimilarity, but it is not clear when that measure was used or the statistical comparison. As another example, the first paragraph of the paper states "Drought changes in bacterial communities link stronger to soil functioning than those in fungal communities" but it appears that that signal to function appears only in the recovery phase.

We have now made it clearer how we calculated resistance and resilience metrics, which we have only used for changes in plant, bacterial, and fungal community composition. As we have now specified in line 127-131, we have calculated Bray-Curtis similarities between drought and control treatments; when drought and control treatments are more similar, they are less affected by drought in their composition. We compare these Bray-Curtis similarities after the drought has been imposed to Bray-Curtis similarities before the drought has been imposed to account for natural variation in community composition that is not caused by the experimental drought.

Thus, and as stated in line 128-131, "When similarities between drought and control communities were significantly lower than those before drought, communities were assumed to be affected by drought and had not recovered. When similarities did not differ significantly, the communities were assumed to be unaffected or had recovered²⁵".

We have now also added a figure to show these Bray-Curtis similarities before and after the drought for bacterial and fungal communities (Fig. 1d and h). Similarly, we use Bray-Curtis similarities between drought and control plant communities at the two-month recovery sampling. Again, a higher similarity between drought and control means a smaller impact of drought and thus a higher resilience (Fig. 6 g,h). We hope that these changes clarify the points raised by the reviewer.

The reviewer is correct that the changes in bacterial communities only link to functioning during the recovery phase; as such, we have now made this explicit in the abstract and throughout the manuscript.

There were two network analysis conducted, correlation networks, which included both positive and negative interactions, and co-occurrence networks, which only consisted of positive correlations. This needs to be spelled out much more clearly. For instance, for correlation networks, a conclusion is that bacterial networks are more stable because of more negative interactions (ln 95) and then, for co-occurrence networks, a conclusion is that bacteria co-occurrence networks (with just positive correlations) were larger and more connected, and so have lower stability than fungal networks (ln 107).

As requested, we now more clearly spell out which analyses were used to test which expectations, and also why. Also, we used the different analyses mentioned by the referee to test different expectations, which are now more clearly spelled out on line 80-95 as follows:

“First, we expected correlations between bacterial OTUs to be stronger overall than those between fungal OTUs, and that a larger proportion of these correlations are negative in fungal networks. We tested these assumptions using all possible correlations in bacterial and fungal networks separately, thus including weak and non-significant interactions. Second, we expected a larger proportion of bacterial than fungal OTUs to be significantly correlated in co-occurrence networks, and these networks to be more connected for bacteria than for fungi. Third, we expected networks of positive correlations to have lower modularity for bacteria than for fungi. We tested these two latter assumptions using co-occurrence networks that only included significant, positive correlations. Positive correlations within fungal and bacterial communities can represent a range of interactions or simply indicate that they are responding in the same way to a change in environmental conditions^{21,22}. Thus, although caution is needed in interpreting them^{22,23}, co-occurrence networks can reveal information on co-oscillation of microbial taxa²¹ and the stability of communities.”

Relatedly, the paper needs more justification for a couple of main methodological analysis decisions. It was not clear why just positive edges were included in the network co-occurrence analysis.

We agree that it was unclear why we only used positive edges in our network analysis. We have now justified our approach in line 88-93:

“Positive correlations within fungal and bacterial communities can represent a range of interactions or simply indicate that they are responding in the same way to a change in environmental conditions^{21,22}. Thus, although caution is needed in interpreting them^{22,23}, co-occurrence networks can reveal information on co-oscillation of microbial taxa²¹ and the stability of communities.”

While we could have included negative interactions, as well as non-linear interactions and three-way interactions, it is much harder to link these properties to network stability because these interactions might not represent true interactions. While co-occurrence networks might equally not indicate ‘real’ interactions between species, regardless of their exact nature, positive interactions do indicate co-oscillation between OTUs and can therefore inform on the stability of these networks. Our

method of using Spearman's rank correlation is, as far as we are aware, the best method available for testing two-species linear ecological relationships (Weiss et al. 2016 ISME). However, we have now included the proportion of negative interactions in the networks consisting of significant interactions and whether these are affected by drought, and now include the following text, line 154-159:

"When considering only significant correlations ($\rho > 0.6$, $P < 0.01$), again we found that fungal networks contained fewer negative correlations than bacterial networks (Chi-square > 10 and $P < 0.001$ for all fungal-bacterial control network pairs). Drought reduced the proportion of negative correlations in bacterial networks at the end of the drought, and in fungal networks after two-months of recovery (Chi-squared = 9.6, $P = 0.002$ and Chi-squared = 9.5, $P = 0.002$, respectively)."

Also, justification for the statistical analysis of the network statistics using each node within a community as a statistically independent point (e.g., Fig 2, 3) is needed. A hierarchical model that takes into account mesocosm-level replication or aggregation to the mesocosm-level may be appropriate.

We understand the reviewer's point. However, one network does not represent one mesocosm; it consists of the correlations of OTU abundances across 36 (drought or control) mesocosms. We chose to construct our networks based on all 36 drought or control mesocosms (rather than create, for example, three replicate networks for only 12 mesocosms) because this gives the most robust correlations (based on 36 observations). Thus, we do not have replicate networks within treatments and it is not possible to nest these node level statistics in networks, and therefore we can't use a nested model. However, while ideally we would have replicate networks, we believe that node-level statistics are robust because the nodes are based on 36 independent observations.

The result starting In 137 confused me, as I was expecting a correlation result and instead several regression stats were listed. Also: what does this statement mean? "While these findings support recent findings that subordinate bacterial taxa respond strongest to precipitation changes, here we show that it is the abundant bacterial taxa that drive bacterial network structure."

We have rewritten this results section to make it clearer, and indeed we are talking about regressions here – this has been corrected. This section now reads (line 180-190):

"We also found that the centrality (betweenness) and connectedness (normalised degree) of bacterial network OTUs was positively related to their relative abundance, but only in drought networks ($R^2 = 0.016$, $P = 0.0002$, $df = 884$ for betweenness in drought late recovery network; $R^2 = 0.037$, $P = 0.0002$, $df = 367$; $R^2 = 0.010$, $P = 0.002$, $df = 884$, for normalised degree in drought early and late recovery networks). In contrast, these properties were strongly positively related to OTU relative abundance in fungal control networks. Before the simulated drought, connectedness was strongly predicted by OTU abundance in both fungal control and drought networks ($R^2 = 0.37$ and $P < 0.0001$, $df = 86$ and $R^2 = 0.22$ and $P < 0.0001$, $df = 111$, respectively), and both centrality and connectedness were strongly predicted by relative abundance in the late recovery control network ($R^2 = 0.49$ and $P < 0.0001$, and $R^2 = 0.24$ and $P < 0.0001$, $df = 101$ for betweenness and normalised degree, respectively)."

We have now changed the results section in accordance with Nature Communications guidelines, and therefore the interpretation of the results the reviewer mentions has moved to the discussion. In lines 175-190 we find that:

- 1) The most abundant bacterial OTUs are also the most central ones in the drought networks (positive relationship between abundance and centrality and betweenness)*
- 2) There is a negative relationship between OTU relative abundance and the P-value of OTU response to drought, so the most abundant OTUs show the strongest response to drought*

Our interpretation of these findings in the discussion now reads (line 265-270):

“The most dominant bacterial taxa were the strongest responders to drought (negative correlation between abundance and P-value of drought response), which is in contrast to previous studies^{31,32}, and we found that these dominant taxa drive network structure (positive correlations between connectedness and betweenness, and abundance). Thus, the most abundant OTUs were driving bacterial, but not fungal, network reorganisation in response to drought^{26,28,29}”

It was interesting that the treatments started with the establishment of four types of plant communities, but the analyses consisted of correlations with *D. glomerata* abundance. Perhaps the authors could explain the original design and their reasoning here a bit more.

*We have now changed the manuscript to Nature Communications format, which allows more space to include experimental details. We have included more information about the experimental design in the main text (line 101-109), the Methods (line 532-571), and the Supplementary Material (Suppl. Table 2). Our reasoning for using *D. glomerata* abundance for explaining drought effects on microbial communities and processes is that drought strongly promoted the abundance of *D. glomerata*, and this increase in *D. glomerata* biomass was the driving force behind drought-induced changes in plant community composition, as stated in line 199-201.*

Lastly, I'd urge the authors to really think about their main conclusions from this paper concerning network stability given the overall finding that there were only weak links between microbial compositional changes and the functions measured. Rather than thinking about resilience in terms of composition, is it that changes in composition lead to resilience in function? One definition of resilience is the ability to reorganize to stabilize function.... Particularly since one of the main conclusions (last sentence in the first paragraph) is the importance to soil function. And were there no relationships between function and network structure?

We agree that there are multiple ways of looking at resistance and resilience. Here, we focus on the stability of bacterial and fungal networks and community composition, and examine whether the drought-induced changes in those communities link to function. Indeed, we only find a link between bacterial communities and functioning during recovery, indicating functional redundancy within those communities. However, we also show that bacterial networks in particular remain strongly altered two months after ending the drought, and that this might have implications for their future response to disturbance.

Unfortunately, it is not possible to link changes in networks directly to changes in functioning because each network consists of correlations between OTUs in 36 mesocosms, while soil functions have been measured per mesocosm.

Comments on writing:

Many very long paragraphs that lose the focus on their topic sentences

Ln 33-37 long sentence, lots of commas.

Ln 60-64, very long sentence.

Ln 79: is recovered the right word here?

Ln 85, comma in a strange place

Ln 105-112 is one long sentence, and tense changes.

We have checked and rewritten large parts of the text, including these sentences highlighted by the reviewer.

Reviewer #2 (Remarks to the Author):

The paper by de Vries and colleagues examines the effect of intense drought on fungal and bacterial communities. In particular they build upon previous work in this area examining network properties. They find bacterial networks are less stable than fungal networks, which likely has implications for ecosystem function. Further, using SEM they link the functions of N₂O flux and denitrification genes to plants. Overall this is an interesting study with some issues that need to be addressed.

General comments

Overall this is a very interesting and well-written paper. It seems that this came from Nature and given that, the lack of detail is understandable. I, and presumably readers, would like to see expanded introduction and discussion that better sets the scene for the data.

As requested, we have now completely rewritten the introduction, and expanded the rationale for our study, in lines 40-98. We have also rewritten the discussion to include more detail, in particular on the functioning (lines 300-315).

My general issue with the study is the delineation between effects of drought and the indirect effects of drought mediated through the plants. The authors sample to 10 cm and presumably sample the rhizosphere of these plants and if these grasses are like most grasses, there should be a lot of root biomass (the authors mention measuring it but don't report it). We see from the data that the bacterial networks are more sensitive than the fungi but there's no way to really say if this was simply due to drought or is an effect of rhizodeposition. Based on extended figure 7e there's a significant effect of drought/recovery on plant community composition and presumably the interactions between plants and microbes. Further, the authors measure geochemical parameters but don't report them anywhere and it would be helpful for determining if the effects of drought on the chemistry, which would be affecting the bacteria. The SEM might be able to get at this idea, but the authors need to tread carefully when discussing this and attribute the changes to both plants and the drought.

We agree with the reviewer that we are measuring both the direct and indirect effects of drought here, and we agree that it is not possible to completely disentangle the direct and indirect effects of drought. However, we use a gradient of different plant

communities, and where the response of microbial properties is affected by plant community composition we can infer that this is because the plant community modifies the effect of drought belowground (and thus an indirect effect of drought). Moreover, and as the reviewer acknowledges, we use structural equation modelling to infer causal relationships and to statistically disentangle direct and indirect effects of drought on soil microbial community properties. We find that our hypothesised causal structure fits the data well: plant communities reduce soil moisture through increased aboveground biomass, which in turn affects microbial communities. We hope that the reviewer agrees that the opposite pattern is not very likely, namely that microbial communities modify soil moisture, which results in higher plant biomass. However, while our hypothesised causal structure is supported by our SEM, we have toned down inference of causality throughout the manuscript.

The reviewer is correct that at the final sampling, soil available C and N, root biomass, and root traits were measured. We have now reported these data in the Supplementary Materials (Suppl. Figures 9 and 10). Because these properties were only measured in the final destructive sampling, they were not included in our repeated measures SEM. We did, however, include these measures in the SEM for the final sampling, to look more in depth at the mechanisms causing changes in microbial community properties. We have now included our a priori model (Suppl. Fig. 4), which did include soil inorganic N and dissolved organic C. However, while these properties did change in response to drought, they did not explain the response of soil microbial communities and hence dropped out of the model (see the final model in Fig. 8).

Throughout the authors could use more explanation of their data and there's certain places they could diver deeper in their interesting results. Further, there are several places where there's no enough information to figure out how the authors did what they did (i.e. CH₄ fluxes, NEE calculations, ect).

We have now rewritten the results section to include more detail, and now include more detail in the discussion on the response of fungal and bacterial communities to drought (line 258-272) and links with functioning (line 300-315). We have also included more detail in the methods, particularly on the climatic conditions (line 519-520 and 532-536), measurement and calculation of gas fluxes (line 545-554), soil and root trait analyses (line 558-571), and qPCR of *nir* and *nosZ* and *amoA* genes (line 573-605).

Abstract

Line 30: the authors don't address negative interactions in this paper. But they probably should.

As we outlined above in response to reviewer 1, co-occurrence networks are easier to interpret than networks containing also negative interactions. While significant positive correlations do not necessarily represent 'true' interactions, they do indicate co-oscillation under changing environmental conditions or over time. Thus, we can infer that larger and more connected networks are less stable, which was the aim of the present study. However, we do agree with the reviewer that the number of negative interactions is also informative. We already included the proportion of negative correlations of all possible correlations (Fig 3 and line 152-154); we now also include the proportion of negative correlations in networks consisting of significant correlations only, and the effect of drought on these proportions in bacterial and fungal networks (line 154-159).

Line 34: I would mention that they are mesocosms, not grasslands.

Done.

Introduction

The introduction, as a whole, needs more detail as it only stands as 1 true paragraph that's bridged between the abstract and the rest of the main text.

We have significantly completely rewritten and expanded the introduction to provide a stronger and more comprehensive background to our study.

Lines 60-64: text like this is very helpful for interpreting the network metrics as what they mean can be difficult to remember.

We have now expanded this section detailing how we assess the different network properties to test our expectations about network stability (line 74-98).

Line 76: N₂O flux is the result of not just denitrification, but also ammonia oxidation and fungal denitrification.

The reviewer is correct – we have changed this sentence to reflect this (line 111-112).

Results/Discussion

Line 77: it would be nice to see the R-values for the PERMANOVA. This would be informative to interpreting the stats in the figures. Also, was drought the only parameter used in the PERMANOVA?

Plant community treatments were also included in the model and we have now reported those statistics in the graph; in addition we have reported R² values in the graph as requested by the referee.

However, it is important to note that plant communities changed significantly over time and as a result of drought (see Fig. 6) and initial dominant species and evenness did not represent plant communities at the time of sampling.

Line 78: an increase of 0.07 in fungal evenness isn't a dramatic change in community evenness, same can be said about the decrease in bacterial evenness.

We agree, and we are now reporting both richness and evenness of bacterial and fungal communities, and how these are affected by drought (Fig. 1 b and f). While the absolute change in evenness is indeed small, the absolute change in richness is more dramatic.

Line 84: It is important to note that fungi sporulate as well. Further, bacterial dormancy would explain a lack of change in community composition, but the authors do see a change for bacteria at all times following drought stress. In addition, I do not buy the argument about osmotic stress killing off bacteria since there's considerable osmotic stress from the drought itself.

We agree with the reviewer and, as such, have now put less emphasis on this result and the mechanisms behind it, since this is not the novel aspect of our study; it has been shown multiple times that bacteria are more affected by drought than fungi. Instead, in the discussion we now focus more on the network properties and their response, and we focus in on specific strongly responding bacterial and fungal taxa.

Line 100: why only positive correlations? I didn't see justification for this and would seem to be an interesting component for this study to identify negatively responding taxa and how they're network metrics respond.

We have now written a more complete justification for why we did which analysis, and why we focus on positive co-occurrence networks, in line 74-98. See also our response above and to reviewer 1:

While significant positive correlations do not necessarily represent 'true' interactions, they do indicate co-oscillation under changing environmental conditions or over time. Thus, we can infer that larger and more connected networks are less stable, which was the aim of the present study. However, we do agree with the reviewer that the number of negative interactions is also informative. We already included the proportion of negative correlations of all possible correlations (Fig 3 and line 152-154); we now also include the proportion of negative correlations in networks consisting of significant correlations only, and the effect of drought on these proportions in bacterial and fungal networks (line 154-159).

Line 113: very cool result!

Thank you!

Lines 118-120: this is cool. It also makes me want to know who is responding here.

We agree that this is cool, but we don't quite understand to who the reviewer is referring here. The text the reviewer refers to reads: "In particular, drought increased the connectedness and centrality of nodes in bacterial networks, while it decreased these properties in fungal networks (Fig. 2)."

Lines 128-129: these r-values are very, very low.

We agree that there is a lot of unexplained variation, however, this is a significant relationship that is based on a large number of observations.

Line 166-167: these results makes me question the main interpretation of the data. Are the plants driving this or is it the drought?

We agree with the referee that our experimental set up does not allow for disentangling the direct and indirect effect of drought. However, we would like to reiterate that our experimental set up included a gradient of plant community treatments that differed in their response to drought, which gives evidence that plant community composition did modify belowground response to drought.

In addition, we do believe that our SEM allows us to at least statistically disentangle the direct and indirect effects of drought. Specifically, in the repeated measures SEM in Fig. 9, soil moisture at the end of the drought is a proxy for the direct effect of drought (plant communities in drought treatments are inactive and not affecting soil moisture content), while after rewetting soil moisture is an integration of the direct effect and the indirect effect of drought via plant communities. However, throughout the manuscript, we have toned down implied causation.

Line 176: The authors, somewhere, need to address why they only targeted denitrification when looking at N₂O flux and not ammonia oxidation. We know that ammonia oxidation can account for a substantial portion of N₂O flux (i.e. Santoro et al. 2011, Science). Further, since fungi were targeted in this study, the authors should discuss fungal denitrification as its only produces N₂O (e.g. Maeda et al. 2015, Scientific Reports).

We agree with the referee and we did measure bacterial and archaeal amoA genes. However, these gene abundances did not link to bacterial or fungal community composition, or to fluxes of N₂O and CO₂ and are therefore not included in our SEM. We have included a figure of these genes now in the Supplementary Materials (Suppl. Fig. 8)

Line 179: this specific plant is only part of the plant biomass in the SEM, right? Also the biomass is aboveground?

Yes. We did measure belowground biomass and traits (see Methods line 563-571) and we have now included a figure on these measures in the Supplementary Information (Suppl. Fig. 9). However, these properties did not explain soil nutrient and C availability, fungal or bacterial community composition, or functional gene abundances, and were thus not included in our SEM.

Lines 181-182: wouldn't this primarily be due to evaporation (direct and from the plant)?

Yes, and we are making this point now in line 217-218: "...indirectly associated with fungal and bacterial community composition through reducing soil moisture content, which was caused by an increase in aboveground biomass and thus evapotranspiration (Fig. 8)."

Line 182-183: was bacterial-N interactions examined? Seems like bacteria should have some correlation.

Yes, we did examine bacterial – soil dissolved organic and inorganic N interactions at the final, destructive sampling. However, there were no significant relationships between these properties, and these links dropped out of the SEM. We have now explicitly stated this in line 220-223:

*"Surprisingly, soil inorganic and organic dissolvable nitrogen and dissolved organic carbon was not associated with bacterial or fungal community composition, despite our observation that the *D. glomerata*-associated increase in aboveground biomass reduced soil inorganic N availability ($R^2 = 0.09$, $P = 0.010$)."*

Lines 184-188: I think this result is particularly exciting and I think it need to come out more here (with discussion of priming and legacy effects) and in the abstract.

The reviewer is referring to this original text: "These findings point to a novel mechanism by which drought can have long-lasting effects on soil microbial communities by promoting the dominance of a fast-growing grass; such responses could potentially be common given that increases in abundance of fast-growing grasses are commonly reported following drought, contributing to the maintenance of grassland productivity"

We have now highlighted this result in the abstract (line 32-35), and dedicated an entire paragraph in the discussion to this (line 273-299). Moreover, we have included more discussion on what these changes might mean for future plant community composition in line 294-299, citing our follow-up study that used soil from the current experiment to assess legacy effects on plant-soil feedback and competitiveness.

In addition, we are now discussing in more detail the implications of increased abundances of genes involved in denitrification in line 300-315, including the possibility that these genes are 'primed' to respond to a second drought.

Line 190: would be great to discuss the potential feedback on ecosystem function.

*See above, we have now expanded on what this might mean for future plant community composition in line 294-299: "While these vegetation-mediated legacy effects on soil microbial communities did not result in long-term altered soil functioning, they were shown in a related study to feedback positively on growth of *D. glomerata*⁵, thereby potentially reinforcing its dominance. As such, changes in soil microbial communities resulting from drought have the potential to have legacy effects on plant community composition, in this case triggering a positive feedback on *D. glomerata* abundance⁵, potentially contributing to long-term altered plant community composition."*

Lines 188-190: this gets at my primary concern with the paper. The direct effects of drought are likely not discernable from the effects of plants here.

As outlined above, we agree with the reviewer that it is not possible to experimentally disentangle the direct and indirect effects of the drought here, and we have toned down our wording to reflect this. However, as also mentioned above, we our experimental set up included a gradient of plant community treatments that differed in their response to drought, which gives evidence that plant community composition did modify belowground response to drought. In addition, we are using SEM to statistically disentangle causative relationships.

Lines 197-198: this seems odd as we'd expect they would. Perhaps diversity would better link this?

We realise that the wording here was confusing and was only referring to the final, late recovery sampling. We have now made it clearer that bacterial communities did link to functioning at the early recovery sampling (one week after the drought). Bacterial and fungal communities didn't link to respiration at the end of drought

presumably because at this point the direct effect of drought on these processes is far stronger than the microbial community-mediated effect.

Line 202-205: it does suggest redundancy only if the function remained the same as in control levels. Since no data is presented, I cannot determine if that's the case.

The reviewer is correct and we did include a figure of the processes in the Supplementary Material to show that all gases had returned to control levels at the final sampling (Suppl. Fig. 8).

Line 208: can you better link these network properties to the fluxes/denitrification genes?

Unfortunately, and as also outlined above in response to reviewer 1, it is not possible to link changes in networks directly to functioning because each network consists of correlations between OTUs in 36 mesocosms, while soil functions have been measured per mesocosm.

Lines 212-214: this could also be due to legacy effects in the soil.

The reviewer is referring to the original text stating: "This suggests that shifts in plant community composition resulting from drought could have long-lasting effects on soil bacterial networks, potentially influencing the ability of aboveground and belowground communities to withstand future disturbances^{3,32}."

We agree with the referee that drought causes legacy effects on the soil. However, we show in our SEMs that these legacy effects are moderated by plant community composition (importantly, we also find no link between soil inorganic N availability and dissolved organic C and microbial communities, see line 220-223).

Methods

Given that this is a drought study it would be nice to know what the temperature profile in the area as well as how much water the control plants received.

We have now included long term climatic data on our experimental site in the Methods (line 519-520), we have included information on rainfall during the experimental drought (line 532-534), and we have included a figure in the Supplementary Material detailing the minimum and maximum daily temperature and daily rainfall over the entire duration of the experiment (Suppl. Fig 12).

Further, there's no mention of how photosynthesis was measured (which is in the SEM)?

We have now included a more detailed description of all our gas flux measurements in the Methods (line 543-554).

While not a requirement, but it would be interesting to get at fungal function with something like FunGuild and see how that changes with drought. Since the

denitrification genes (typically, since I don't know what primers were used) quantified are bacterial and don't target fungal nirK.

We agree this would be an interesting addition though feel this is beyond the current manuscript focus. We note that a similar tool for assigning putative functional traits is not available for bacteria, and so exploring this issue would likely need to be done in a separate fungal focussed article.

It is correct that fungal nirK is not targeted in our qPCR, see also Supplementary Table 3.

Line 222-224: Did the plants receive any additional watering or was it just reliant on rain?

In dry weeks the mesocosms did receive extra watering; we have now added text to the Methods detailing this (line 534-536):

"All pots, and control treatments during the drought, were supplemented with 700ml of additional water (equivalent to 0.5mm of rainfall) every two days in dry weeks."

Lines 224-226: I know little of the plants in the area of the world, but are these common plants? It would be helpful to know the rationale as to why the plants were selected.

We have now included more information on the plants used in our experiment, in the main text (line 101-109) as well as in the Methods (line 520-527) and we have added table in the Supplementary Material with the plant community treatments (Suppl. Table 2).

Line 231: are the rain covers just little roofs? Or are they boxes? Also, do these effect the temperature in the area?

Yes, they are transparent roofs of 1x1m. The roofs did affect soil temperature at the end of the drought; however, this might also be caused by the reduced soil moisture content (see Suppl. Fig. 11).

Lines 235-236: is there a reason that grazing was simulated three times throughout the experiment? I know grazing is a stress plants have to deal with, however since the authors are looking at soil (likely including rhizosphere) the changes in root exudates can themselves effect composition of microbes. It would be helpful to know, in relation to the sampling dates, when the plants were clipped.

We have now included more detail on when the plants were clipped in line 537-538. Grazing was simulated because the grasslands our mesocosms represent are typically grazed or cut as part of the normal management.

Lines 239-241: were the samples taken to be representative of all the plants in the pot? Were there roots in the samples? If so were they removed? The roots aspect of my question is important due to the effects of rhizodeposits on the communities.

Yes, we took a composite sample of randomly distributed cores from each pot, see text in line 541-543. Roots were removed and samples were sieved to 2mm.

Lines 253-256: please list the primers used. It is hard to evaluate how encapsulating the qPCR results are of denitrifying organisms. Further, what was used for standards? How was the DNA quantified? Was the DNA normalized for each reaction?

A supplementary table has been added with the primers and thermal cycling conditions (Suppl. Table 3). The primers and DNA was quantified using Qbit and we used 5 ng DNA per reaction. For each of the standards, we used a linearized plasmid with a cloned gene fragment obtained from pure cultures. We have expanded the description of the qPCR assays in the methods section.

Line 274: 18S?

This has now been removed (line 615).

Lines 282-283: where was sequencing performed?

The sequencing was performed in the laboratory at CEH Wallingford. We do not feel specifically stating this information would add to the manuscript.

Line 287: since the authors are comparing fungal and bacterial networks it would be nice to have the sequences processed in the same manner and have that better delineated in the methods.

Where possible this was done, though we note that the two different marker genes ultimately require different processing steps. We have amended the text as follows (line 637-641):

“The fungal ITS sequences were analysed using PIPITS⁵⁷ with default parameters as outlined in the citation. Briefly this involved quality filtering and 97% clustering of the ITS2 region as indicated above for the 16S processing, using the UNITE database for chimera removal and taxonomic identification of representative OTUs.”

Lines 296-298: I have to question the efficacy of using a de novo cluster for fungal ITS reads. Given the range of sequence length (~75bp according to the bioanalyzer data in the text) the diversity estimates (both beta and alpha) would likely be affected, as well as the networks. While it shouldn't change the results dramatically, I would recommend doing a closed reference pick against UNITE to assuage the worry that OTUs are being improperly called because of the differences in sequence length, instead of biology.

This is ultimately a matter of personal preference - use de novo clustering to capture as much diversity as possible albeit with some largely unquantified limitations; or use closed reference clustering to only capture those sequences present in the UNITE database. We chose the former, since recent publications have demonstrated that de novo clustering outperforms reference based picking:

Westcott, S. L., & Schloss, P. D. (2015). *De novo clustering methods outperform reference-based methods for assigning 16S rRNA gene sequences to operational taxonomic units*. *PeerJ*, 3, e1487. <http://doi.org/10.7717/peerj.1487>

Line 300: 'OTU' not out. Also, while I don't agree with the discussion on rarefaction, there's issues with rarefaction (see papers by McMurdie & Holmes). Have the authors tried raw abundances for their data? Or robust rarefaction (i.e. 10,000 resamples)?

We are aware of the issues with rarefaction as reported by McMurdie & Holmes, though we do not feel these issues are pertinent to our analyses. Firstly, rarefaction can add random noise potentially obscuring significant findings. We argue that the drought treatments imposed here are clearly strong enough to drive large effect sizes (differences between treatments over time) and so inflation of error is clearly not an issue given the large differences between treatments. Secondly much of the issues with rarefaction pertain to accurate inclusion of rare taxa into the analyses. Rare taxa are unlikely to be important determinants of any of the analyses we present:

- 1. the ordination results and significance test based on Bray Curtis distances in our experience are weighted towards detecting change in abundant taxa*
- 2. and we have filtered out rare taxa by only using OTUs that occur in a minimum of 8 experimental units, thus removing spurious correlations (as in Shi et al. 2016).*

Thus, we feel in our case the benefits of rarefying to standardise read numbers outweigh the costs— particularly for the network analyses.

*Shi, S. et al. The interconnected rhizosphere: High network complexity dominates rhizosphere assemblages. *Ecol. Lett.* **19**, 926-936, doi:10.1111/ele.12630 (2016).*

Should the R library be vegan?

We thank the reviewer for spotting this and this has now been changed to vegan (line 647)

Line 323: what methodology was used to determine significance and what alpha?

We deliberately chose to not only focus on strong interactions and thus we included all correlations with a $\rho > 0.6$. Significance was tested as in Barberan et al. 2012, i.e. interactions were considered significant if $P < 0.01$. This has now been specified in the text (line 665-668).

Line 323-324: seems like this sentence is missing something at the end. Please revise.

This sentence has been rewritten (line 665-668):

“Interactions consisted of Spearman’s rank correlations and co-occurrence networks were constructed using only significant correlations ($P < 0.01$ as in Barberan et al.²¹) of $\rho > 0.6$; this cut-off was chosen to include a range of interactions strengths (not only strong interactions).”

Line 330: it wasn't mentioned in the text at all or apparent in the SEM figure, so I may be wrong here and if I'm wrong, the authors need to expand on the SEM methods. But why do the SEM only show plants influencing the environment and not the other way around? Further, why aren't the C/reactive N included in the SEM? Both of these are likely to be affecting plants.

We have now included our hypothesised a priori models, including justification, in the Supplementary Material (Suppl. Fig. 4 and 5). As also stated above, our experimental design included a range of plant communities to test the hypothesis that plant community composition alters aboveground and belowground responses to drought. While we completely agree that there are feedbacks between microbial communities and plant communities (see also our discussion in line 294-299 and the response to an earlier comment above), our hypothesis is that plant communities drive belowground responses. While it would be interesting to test the feedback of changes in microbial communities here, it is not possible to include feedback loops in SEMs. Therefore, we now highlight the result of a follow-up study that we did with soil from this experiment in line 294-299, which shows that the changes in microbial communities that we report here do indeed have implications for future plant community composition.

As now explicitly stated, we did include DOC and inorganic N in the SEM of the final sampling (Suppl. Fig. 4). We did not analyse all soil properties at each sampling, see Methods line 554-571); however, these did not explain fungal and bacterial community composition (line 220-223).

Further, since the authors make mention of plants responding to drought, the authors should consider including plant metrics (species, biomass) in the SEM. They can include plant species, perhaps as who's the dominant member at a given time, as a metric in the model as done recently in Nature Communications by Bowen et al. (2017).

*Both our SEMs explicitly include plant community metrics. The repeated measures SEM (Suppl. Fig. 5 and Fig. 9) includes PCA scores of plant community composition, while the SEM of the final sampling also includes biomass of *D. glomerata* and total aboveground biomass (Fig 8 and Suppl. Fig. 4).*

Finally, why did the authors use the ratio of nirS/nirK and nosZ1/nosZ2? What do these ratios truly tell you about the system? It seems that the ratio just indicates the balance between the different types of denitrifiers (which doesn't correlate to N₂O emission potential) as well as the potential for N₂O reduction between canonical denitrifiers and N₂O scavengers. I would suggest that the authors adopt an approach akin to that of Jones et al., (2013), Nature Climate Change. These authors linked N₂O flux to the 4 genes the authors here quantified and that would be better suited to their SEM.

We agree with the reviewer and we have now changed our SEM to include the total abundance of nir genes (nirS+nirK), which would link to increased N₂O production, and the total of nosZ genes (nosZI+nosZII) which would link to reduced N₂O fluxes. See also our a priori model in Suppl. Fig. 5.

The other thing that the SEM doesn't seem to take into account is the link between

gas flux (in particular CO₂ which the plants will be making plenty of) and plant composition. This needs to be addressed somewhere in the model or text.

We did include the link between plant community composition and CO₂ flux in our a priori model (see Suppl. Fig 5), but this link is never significant. Plant community composition does affect CO₂ production indirectly through its effect on photosynthesis (see Fig. 9).

One last thing, wouldn't bacterial/fungal richness be more appropriate here in lieu of PCoA scores?

No – PcoA scores capture more changes in community composition than richness alone and are thus a much more sensitive measure to use in the SEM.

References

There are some inconsistencies in the references that should be resolved.

We have gone through the references and these inconsistencies should be sorted now.

Data availability

Have the authors deposited the data and the associated metadata in a place like MG-RAST or the NCBI SRA? And while not mandatory by any means, I would enthusiastically suggest the authors make their code available on GitHub (or equivalent site) so that others can follow the workflow, in particular the network analyses.

We agree that it is important to make data and code available. Therefore, OTU tables and outputs of the indicator analysis are provided as supplementary files and raw sequences and code are available on request. Note that the code used for the networks analysis is already accessible on github, as stated in the Methods (line 664-665)

.

Figures

In general, the text of the figures are far too small and could use a size increase.

We originally made the text in the figures this size in accordance with Nature guidelines. We have now enlarged the text in all figures, hopefully making them easier to read.

Further, I don't like connecting the dots in the dot plot figures. It implies that there's no dynamic in between and I don't think we can make that assumption here.

We respectfully disagree with the reviewer here. While we acknowledge that there might be dynamics in between sampling dates, surely this is the case for any measurement that ins not continuous. We believe that connecting the dots improves the clarity of the figures and makes it clear that the data represent time series.

Figure 1: These networks are tiny and are hard to make out the individual nodes. I'm

not sure if anything can be done, but just an observation. It would also be nice to note the significance alpha value for the networks in the legend.

We have now tried to make all figures and the text within them larger, and hopefully easier to read. We have kept information on the significance of networks in the Methods and in Supplementary Table 1.

Figure 2: this is a personal preference, but the yellow-orange used in the boxplot is harsh. Also what do the boxes and lines mean? I assume quartiles and median but would be nice to know for sure.

We have added explanation of the boxes and lines to the figure legend.

Figure 3: how are you defining abundance of the plant? The number of shoots or leaves? What do the grey point mean, no correlation? Non-significant correlation?

*Abundance is measured by aboveground dry biomass. We have now specified in the legend that indeed, grey points indicate no significant correlations between OTUs and *D. glomerata* biomass.*

Extended figure 1: the meaning of the grey bar in C-F isn't in the legend. I would also suggest perhaps using Shannon Diversity for C and D, overall richness isn't ideal for microbes.

We have included a statement on the meaning of the grey bar (duration of the drought) in the legend. We also believe that it is more informative to show the separate components of Shannon diversity (evenness and richness).

Extended figure 2: maybe this was just my copy, but this figure was a mess and I couldn't not figure out what was going on except for in the first panel. Also, given how tiny these are is there a better way to represent this taxonomic data? Bar chart maybe?

We have now increased the size of this figure, but note this is a Supplementary Figure that is best viewed in colour on screen.

Extended figure 7 (now fig 6): no mention of how the beta diversity was calculated in the methods section. What does the grey shading refer to in g and h? Please put that information in the legend.

We are not entirely sure which beta diversity the reviewer is referring to? If the reviewer is referring Bray-Curtis similarities, the calculation of these is detailed in line 127-131. We have now included an explanation of the grey shading in the legend (now Figure 6).

Extended figure 8: CH₄ fluxes are not mentioned anywhere in the text, please add to the text (results, methods) or remove panel e. I assume the bars are SE and dots are means? How was NEE calculated? No mention of that in the text.

We have now removed CH₄ from the figure as this measure showed large variation and, as a consequence was not affected by drought and not explained by plant or microbial properties. We have now specified that dots are means and bars are SE. We have also included more detail on how NEE was calculated in the Methods (line 543-554)

Reviewer #3 (Remarks to the Author):

The authors present an interesting study on a highly relevant topic. They examine the effect of a climate change effect, extreme drought events, and test how plants communities as well as bacterial and fungal communities respond to drought and how they recover from it over the course of two years. They created and experimental set-up with plant communities that did or did not receive a drought treatment by rain shelters. The authors sampled prior to, during and two times after drought. From soil sample sequencing data they monitor the change in network structure of bacterial and fungal soil communities using co-occurrence networks and community composition (with NMDS) by looking at taxa richness, evenness, resilience, resistance, change, overlapping taxa, most responsive taxa to treatment with the use of Mantel tests, changes with similarity and dissimilarity indexes with the use of PCoA. Next to that they have measured respiration (CO₂-production) and N₂O-emissions from the system and related that to the gene expression of nirS/nirK and nosZ/nosZI to determine the genetic potential for denitrification and N₂O reduction and the nitrous oxide reductase with Q-PCR. Additional, they performed a structural equation model to indicate cause and effect. They challenged several expectations. First, they expected bacterial networks to show more destabilizing properties, such as high connectance and low modularity, than fungi. Second, they also expected fungal communities to respond more in the recovery phase as changes in plant community structure and plant growth could be associated with the microbial recovery, but as fungi get the carbon first, they are expected to be most responsive. Third they expected drought-induced changes in fungal and bacterial networks to influence soil functioning. The authors confirmed hypothesis one with networks. My major concern with these networks is that also in the control networks there is also a large difference. And although it seems the networks are tested against random networks, I am wondering to what extent the differences in the networks both for the controls and the drought treatment is also caused by seasonal effects as this experiment was monitored over 2 years.

We agree with the reviewer that these networks change over time. However, we want to make it clear that we did not sample these networks over two growing seasons, as the referee implies – all four samplings were done in the second growing season, which is when the drought was applied. But more importantly, we compare drought networks to control networks at the same time point, rather than to previous networks. It has indeed been shown before that bacterial networks show strong patterns over the growing season (Shi et al. 2016).

Shi, S. et al. The interconnected rhizosphere: High network complexity dominates rhizosphere assemblages. Ecol. Lett. 19, 926-936, doi:10.1111/ele.12630 (2016).

Drought stimulated *D. glomerata* in dominating into the mesocosms and that change seemed to influence the bacterial community most. This demonstrates hypothesis two is not supported. Although the authors performed a structural equation model to separate cause and effects, there seems to be some circularity in the reasoning. It is a bit of a chicken and egg discussion as soil communities influence plant

communities already before drought is applied, then during and after drought plant community changes, but it is hard to determine if drought is causing a direct effect on plant growth, or changes the microbial community first which is then causing plant community shifts. Of course it could also be the other way around, but I do not get why the authors are so sure of that.

We agree with the reviewer that our experimental set up does not allow to fully disentangle whether changes in plant communities affect microbial communities or the other way around. However, and as stated in response to reviewer 2, our experimental design included a range of experimentally manipulated plant communities to test the hypothesis that plant community composition alters aboveground and belowground response to drought. While we completely agree that there are feedbacks between microbial communities and plant communities (see also our discussion in line 294-299), our hypothesis is that plant communities drive belowground responses, which was supported by our SEMs. See also our response to reviewer 2 that it is unlikely that microbial communities modify soil moisture, resulting in higher plant biomass. However, we have toned down our wording to infer causality throughout the manuscript.

Hypothesis three seems to be confirmed with a stronger effect from bacteria to gene expression. I did not see that immediately from the figures and I am just wondering if this is not slightly biased as I guess gene expression levels of nirS/nirK and nosZ/nosZI is more pronounced in bacteria over fungi, and then it is not surprising that they are more expressed in bacteria. But I might be wrong here, as I am not a specialist in the gene expression of these genes.

There is no known fungal genome containing nirS, nosZI or nosZII. There are fungi harbouring nirK, but they are not targeted with the primers we used (See Bonilla-Rosso, G., Wittorf, L., Jones, C.M. and Hallin, S. 2016. Design and evaluation of primers targeting genes encoding NO-forming nitrate reductases: implications for ecological inference of denitrifying communities. Scientific Reports, 6:39208.)

Minor points:

- Extended data figure 2 is extremely hard to read and understand. Is there a Mantel test behind this indicating the most responsive OTU's in the outer ring? There are also some corners missing in the figure.

We have improved the quality of the figure. As indicated in the text (line 651-660), the significance of drought responsive indicators was determined using Indicator Species analyses (Dufrene & Legendre).

- I am not sure how the community structure is one to one linked to gene-expression.

We are not entirely sure what the referee is referring to, but the link between bacterial and fungal communities and functional gene abundances, as included in our SEM, assumes that changes in fungal or bacterial community composition will predict changes in functional gene abundance, which was supported by our SEM.

REVIEWERS' COMMENTS:

Reviewer #2 (Remarks to the Author):

The authors have satisfactorily addressed my comments and I am very excited about the results of their work. It represents a substantial increase in our knowledge about soils and their response to drought.

Reviewer #3 (Remarks to the Author):

I carefully read the reports of the other reviewers and how the authors responded and how they subsequently edited the manuscript. I was impressed by how much the manuscript had improved. I found the rebuttal very strong, very thorough. They took away the doubts I had about some points. I think the authors did a good job, it is now a very nice paper based on a robust body of work. It is ready for publication.